# ComPEFT: Compression for Communicating Parameter Efficient Updates via Sparsification and Quantization

Prateek Yadav[1]   Leshem Choshen[2,3]   Colin Raffel [4,5]   Mohit Bansal[1]
[1] UNC-Chapel Hill, [2] MIT, [3] MIT-IBM Watson AI Lab,
[4] University of Toronto, [5] Vector Institue
**Correspondence Email: {praty@cs.unc.edu}**

Reviewed on OpenReview: `https://openreview.net/forum?id=CovLQwu611`

## Abstract

Parameter-efficient fine-tuning (PEFT) enables creation of specialized language models for diverse tasks, resulting in numerous expert modules. In many practical use cases, these expert PEFT modules are integrated into a single model that answers arbitrary queries by routing queries to different experts. However, only a few experts can be kept in GPU memory due to memory constraints. Consequently, expert modules are frequently loaded and offloaded between CPU/GPU memory or disk storage. This frequent swapping dramatically increases communication overhead, leading unacceptable latency and degrading user experience. The large size of modern PEFT modules further exacerbates this latency. For example, QLoRA experts for `65B LLaMA` are `3.2GB`, making swapping a major communication bottleneck, particularly in memory-constrained environments. To address these issues, we present `ComPEFT` (compressed PEFT), a novel method for compressing fine-tuning residuals (task vectors) of PEFT models. Reducing expert PEFT module size effectively addresses both memory and communication limitations, facilitating faster swapping and enabling a higher density of experts within a given memory footprint. `ComPEFT` employs sparsification and ternary quantization to reduce PEFT module size without any additional training while preserving or enhancing model performance. Extensive evaluation across `T5`, `T0`, and `LLaMA`-based models with $200M - 65B$ parameters, `ComPEFT` achieves compression ratios of `8x − 50x`. Specifically, we show that `ComPEFT` improves with scale – stronger models exhibit higher compressibility and better performance. We show `ComPEFT` applied to `LLaMA − 65B` outperforms QLoRA by $4.16\%$ on MMLU with a $26x$ storage size reduction. Additionally, compressed experts produced by `ComPEFT` maintain few-shot compositional generalization capabilities, facilitate efficient communication and computation, and exhibit enhanced performance when merged. Lastly, we provide an analysis of different method components, compare `ComPEFT` with other PEFT methods, and test its efficacy for compressing full finetuning residual.[1]

## 1 Introduction

Parameter-efficient fine-tuning (PEFT) (Houlsby et al., 2019; Karimi Mahabadi et al., 2021) methods like LoRA (Hu et al., 2021) and (IA)³ (Liu et al., 2022) efficiently adapt language models by learning only a few new parameters. QLoRA (Dettmers et al., 2023) further reduces memory needs by using 4-bit quantization for the base model. This combined efficiency has fueled a surge in specialized models for diverse tasks such as multimodal understanding (Zhang et al., 2023), multilingual applications (Yang et al., 2023), and expert systems for math (Luo et al., 2023a) or coding (Luo et al., 2023b). Platforms like HuggingFace Hub (Wolf et al., 2019) now host a rapidly growing collection of these expert PEFT models.

---

[1]Code is available at `https://github.com/prateeky2806/ComPEFT`.

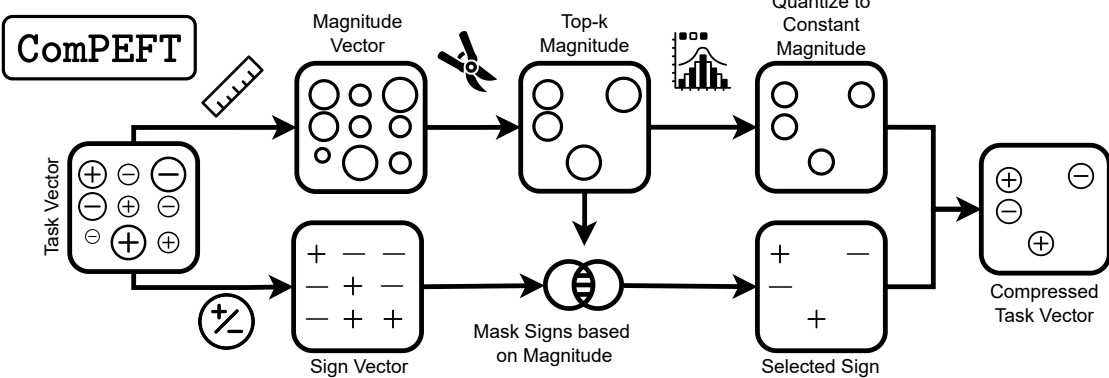

Figure 1: `ComPEFT` without any additional training compresses PEFT modules while preserving or enhancing model performance.

Serving these expert PEFT models has different strategies. LoRA, for instance, allows merging expert PEFT modules into the base model for single-expert low-latency inference and proposes expert swapping for sequential multi-expert serving. While efficient for single and sequential multi-expert serving, these methods become slow for concurrent multi-expert serving because swapping reduces throughput and increases latency (Sheng et al., 2023). Furthermore, LoRA's approaches don't fully utilize available GPU memory for a larger number of experts. For efficient high-throughput concurrent serving, separating base model and adapter computations is crucial, as multiple experts can then share the base.

This enables efficient base model batching, but directly batching expert PEFT modules remains challenging. Serving numerous experts demands efficient memory management. Limited GPU memory necessitates storing experts off-GPU and dynamically fetching them when needed. This dynamic loading of large expert modules leads to communication overhead and unacceptable latency, degrading user experience (Sheng et al., 2023). Communication bottlenecks extend beyond concurrent multi-expert serving. Techniques like Model merging (Ilharco et al., 2023; Yadav et al., 2023), Model MoErging (Yadav et al., 2024), and compositional generalization (CG) (Huang et al., 2023; Muqeeth et al., 2024) also require dynamically retrieving expert PEFT modules from cloud/disk/cache into GPU memory based on input queries to dynamically merge or route through experts for improved generalization. Consequently, these methods also face communication challenges.

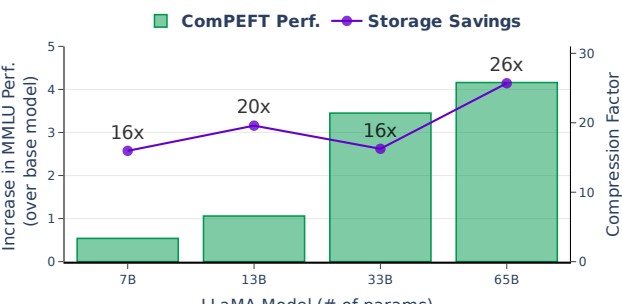

Figure 2: `ComPEFT` **improves performance with larger base models while compressing models significantly. Left Axis:** Improvement of MMLU performance over the corresponding base model. **Right Axis:** Compression factor achieved by `ComPEFT` compared to the original checkpoint.

For example, a `3.2 GB` QLoRA adapter for `LLaMA − 65B` (comparable to a full T5-Large model (Raffel et al., 2020b)) can make frequent swapping a bottleneck, especially under memory constraints. Therefore reducing expert PEFT module size solves both memory and communication issues by facilitating both faster swapping and increased expert density within a given memory footprint.

To address these issues, we introduce our `ComPEFT` (compressed PEFT) method that compresses fine-tuning residuals – i.e., task vectors – by exploiting their inherent redundancies (Yadav et al., 2023). The task vectors represent the learned changes to the model's parameters during fine-tuning for a specific task. `ComPEFT` achieves this compression through a two-step process. First, it applies sparsification, aggressively sets a large portion of the values within the PEFT task vector to zero. This step is based on the observation that many values in task vectors are close to zero and contribute minimally to the expert's behavior (Yadav et al., 2023). Second, for the remaining non-zero values, `ComPEFT` employs ternary quantization. Instead of storing these

values at full precision, it represents their magnitudes using a single, shared scalar constant, and their signs $(+1, -1, \text{or } 0)$. This results in task vectors with sparse ternary weights, drastically reducing their size (see Figure 1). `ComPEFT` shares similarities with the Sparse Ternary Compression (STC, Sattler et al., 2019a) method used in federated learning, however, there are notable differences. Unlike STC, `ComPEFT` retains high-performance without the need for additional training. This is in stark contrast to directly applying STC to task vectors, which we find leads to performance degradation. Remarkably, `ComPEFT` can often restore and even surpass the original fine-tuned performance by carefully calibrating the magnitude of the shared scalar constant used in ternary quantization. Additionally, we demonstrate this effectiveness not only for PEFT modules but also for fully fine-tuned models. Moreover, we observe a beneficial trend that the optimal magnitude of this scalar constant becomes consistent across different tasks for larger models ($\geq$ `13B`). This eliminates the need for task-specific tuning and simplifies the practical deployment of `ComPEFT` at scale, further facilitating reduced latency during model serving. Finally, the `ComPEFT` compression enables more efficient operations on task vectors that can facilitate faster merging of models and compute their similarity.

We perform comprehensive experiments for `ComPEFT` to evaluate: (1) the performance of the compressed model on its original tasks, (2) the number of bits needed to store the models, (3) the mergeability and composability of the compressed checkpoints, and (4) how `ComPEFT` compares to other existing PEFT methods. We performed experiments with `T5` (Raffel et al., 2020a), `T0` (Sanh et al., 2021a), `LLaMA` (Touvron et al., 2023a), and `LLaMA2` (Touvron et al., 2023b) as the base models with model sizes ranging from `200M` $-$ `70B` parameters. We found that in most cases `ComPEFT` can provide compression of `8x` $-$ `50x` (compared to `16` $-$ `bit` precision checkpoints) while performing similarly or better than the uncompressed models. Additionally, we note a surprising finding that as the base model gets bigger, their task vectors become more compressible and these compressed checkpoints significantly outperform the original uncompressed checkpoints. Specifically, as shown in Figure 2, `ComPEFT` leads to an improvement of $0.54\%$, $1.06\%$, $3.44\%$, and $4.16\%$ on MMLU for QLoRA trained on `LLaMA 7B`, `13B`, `33B`, and `65B` parameter models, respectively, while compressing model by `16x` $-$ `26x`. Beyond performance and size, we demonstrate that `ComPEFT` provides order-of-magnitude reductions in model transmission and loading latency, directly addressing communication bottlenecks in expert model serving. In addition, we show that (1) the compressed models from `ComPEFT` lead to better-merged models; (2) for few-shot compositional generalization (CG) (Huang et al., 2023), `ComPEFT` checkpoints lead to similar performance on BBH (Suzgun et al., 2022); (3) `ComPEFT` applied to LoRA and (IA)$^3$ is Pareto-optimal in terms of storage costs vs. performance compared to a wide range of existing; (4) the importance of the components of `ComPEFT` in an ablation study, and (5) the effect of sparsity and scaling on performance. We further show that `ComPEFT`'s benefits extend beyond PEFT, effectively compressing fully fine-tuned models with minimal degradation. Our results and analysis establish `ComPEFT` as an effective method for compressing task vectors. In summary, our contributions are:

1. `ComPEFT` demonstrates that even efficient PEFT modules can be drastically compressed (`8x` $-$ `50x`) via sparsification and quantization without performance loss, suggesting PEFT modules contain significant redundancy.

2. The reduced size of `ComPEFT` checkpoints (`8x` $-$ `50x` smaller) enables serving larger models or significantly more expert PEFT modules given fixed resources.

3. `ComPEFT`'s smaller size reduces communication overhead during dynamic retrieval and loading/offloading, leading to improved latency in practical serving systems.

## 2  `ComPEFT`: Compression for Communicating Parameter Efficient Updates via Sparsification and Quantization

As discussed in the introduction (§1), serving numerous expert PEFT modules suffer from communication and memory bottlenecks which can be alleviated by compressing the experts. This section details `ComPEFT`, our method to compress experts by targeting their fine-tuning residuals. Given a pre-trained model like `LLaMA` (Touvron et al., 2023a) or `T5` (Raffel et al., 2020b), we can create an expert model for specific task $t$ by either finetuning all model parameters or using a parameter-efficient fine-tuning (PEFT) approach such as (IA)$^3$ (Liu et al., 2022) or LoRA (Hu et al., 2021). In both scenarios, we represent the trainable parameters

as $\theta$, initialized as $\theta_{\texttt{init}}$, which, upon fine-tuning, become $\theta_{\texttt{ft}}$. This work assumes access to the initial model parameters $\theta_{\texttt{init}}$ and the fine-tuned model parameters, $\theta_{\texttt{ft}}$. Our work focuses on two key objectives: (1) to achieve extreme compression of parameter updates for efficient and low-latency communication of expert models, and (2) to gain insights into the inherent compressibility of these updates, suggesting a lower intrinsic dimensionality of learned task-specific knowledge.

For a given task $t$, we first represent the parameter updates from fine-tuning as a task vector $\tau_{\texttt{t}} = \theta_{\texttt{ft}} - \theta_{\texttt{init}}$. This task vector encapsulates the changes applied to the base model parameters to specialize it for task $t$. To effectively compress these task vectors, we decompose each $\tau_{\texttt{t}}$ into direction and magnitude components. This decomposition allows us to treat the direction of parameter updates and their magnitude separately, enabling us to apply distinct compression strategies optimized for each component. We decompose the task vector $\tau_{\texttt{t}}$ into a direction (sign) vector $\gamma_{\texttt{t}} \in \mathbb{R}^d$ and a magnitude vector $\mu_{\texttt{t}} \in \mathbb{R}^d$. Formally, the direction vector $\gamma_{\texttt{t}} = \text{sgn}(\tau_{\texttt{t}})$ captures the sign of each parameter (+1, 0 or -1), while the magnitude vector $\mu_{\texttt{t}}$ is $\mu_{\texttt{t}} = |\tau_{\texttt{t}}|$ captures the absolute magnitude. This decomposition allows us to express the task vector as the Hadamard product: $\tau_{\texttt{t}} = \gamma_{\texttt{t}} \odot \mu_{\texttt{t}}$. Based on the intuition from Yadav et al. (2023), that the direction of parameter updates is crucial for task adaptation, while lower magnitude updates are redundant. `ComPEFT` achieves high compression by sparsifying the direction vector $\gamma_{\texttt{t}}$ and quantizing the magnitude vector $\mu_{\texttt{t}}$ to a single scalar.

### 2.1 Steps in `ComPEFT`

To reconstruct an expert model for task $t$, we only need to communicate the compressed update over the base pre-trained model, which is represented by the task vector $\tau_{\texttt{t}}$. As described earlier, we decompose this task vector into a direction vector $\gamma_{\texttt{t}}$ and a magnitude vector $\mu_{\texttt{t}}$. Given this decomposition, `ComPEFT` compresses the task vector through two key steps of sparsification and quantization. Refer to Algorithm 1 and Figure 1.

---

**Algorithm 1** `ComPEFT` Compression Procedure.

**Input:** Task vector $\tau_{\texttt{t}}$, $k$, and a scaling value $\alpha$.
**Output:** Compressed task vector $\tilde{\tau}_{\texttt{t}}$
$\gamma_{\texttt{t}} \leftarrow sgn(\tau_{\texttt{t}})$
$\mu_{\texttt{t}} \leftarrow |\tau_{\texttt{t}}|$
▷ Step 1: Sparsify.
$\tilde{\tau}_{\texttt{t}} \leftarrow \text{keep\_topk\_reset\_rest\_to\_zero}(\gamma_{\texttt{t}}, \mu_{\texttt{t}}, k)$
▷ Step 2: Quantize Magnitudes to scalar.
$\tilde{\tau}_{\texttt{t}} = \alpha * \sigma(\tau_{\texttt{t}}) * \tilde{\gamma}_{\texttt{t}}$
**return** $\tilde{\tau}_{\texttt{t}}$

---

1. **Sparsify:** We sparsify the direction vector $\gamma_{\texttt{t}}$ by retaining only the signs of the parameters corresponding to the top-$k$% largest magnitudes in $\mu_{\texttt{t}}$, and setting the signs of the remaining $(1-k)$% parameters to zero. Following Yadav et al. (2023), we select the top-$k$% parameters based on their magnitude in $\mu_{\texttt{t}}$ as larger magnitude updates generally represent more significant parameter changes learned during fine-tuning. By preserving the signs of these largest magnitude updates and zeroing out the rest, we aim to retain the most critical directional information for each task. Formally, the sparsified direction vector, $\tilde{\gamma}_{\texttt{t}} = \gamma_{\texttt{t}} \odot \text{top-}k(\mu_{\texttt{t}})$, where top-$k(.)$ is applied elementwise and returns 1 for indices with the $\texttt{top} - \texttt{k}$% magnitude values and 0 otherwise. The parameter $k$ is referred to as the "*density*", and $1-k$ as the *sparsity*.

2. **Quantize Magnitudes:** We then quantize the magnitude vector $\mu_{\texttt{t}}$ to a single scalar value. Specifically, we define the compressed task vector $\tilde{\tau}_{\texttt{t}} \in \mathbb{R}^d$ as $\tilde{\tau}_{\texttt{t}} = \alpha * \sigma(\tau_{\texttt{t}}) * \tilde{\gamma}_{\texttt{t}}$. Here, $\sigma(\tau_{\texttt{t}}) \in \mathbb{R}$ is the standard deviation of the original task vector $\tau_{\texttt{t}}$, and $\alpha \in \mathbb{R}$ is a scaling hyper-parameter. We utilize the standard deviation of the original task vector as a scaling factor to normalize the magnitude, which helps to make the optimal $\alpha$ value more consistent across different tasks and models. Refer to Appendix B.5 for more discussion. The scaling factor $\alpha$ is then chosen by evaluating performance on a small validation set; importantly, $\alpha$ is the only parameter tuned during this process. We observe that this simple scalar scaling is sufficient to effectively mitigate any performance loss from sparsification and ternary quantization. This contrasts with many model pruning methods that require computationally expensive retraining after sparsification to recover performance.

### 2.2 Efficient Storage of `ComPEFT` Models

`ComPEFT`'s compression strategy directly addresses the communication bottleneck and latency concerns highlighted in §1 by significantly reducing the storage footprint of expert PEFT modules. This section

details the storage efficiency gains and discusses practical encoding schemes for efficient communication and computation.

**Entropy of the Sparsified Task Vector.** A typical task vector $\tau_{\mathtt{t}}$ in bfloat16 or fp16 format requires $16 * \mathtt{d}$ `bits` for memory/storage. Assuming uniform value distribution, its entropy is also $\mathbb{H}_{\mathtt{dense}} = 16 * d$ `bits`. ComPEFT, however, represents the compressed task vector $\tilde{\tau}_{\mathtt{t}}$ using a sparse ternary sign vector (values $\in \{-1, 0, +1\}$) and a single 16-bit scalar value ($\alpha * \sigma(\tau_{\mathtt{t}}) \in \mathbb{R}$). Assuming the signs of the nonzero entries of $\tilde{\tau}_{\mathtt{t}}$ are uniformly distributed, the ternarization step reduces the entropy of the update to $\mathbb{H}_{\mathtt{ComPEFT}} = -((1-k)\log_2(1-k) + k\log_2(\frac{k}{2})) * d + 16$ `bits`, where $k$ is the density of the update. At a density level of $k = 0.05$, the resultant update has 95% of the values as 0 and the entropy is $0.34 * d + 16$ `bits`. Hence, with a perfect encoding-decoding scheme and 95% sparsity, our ComPEFT can reduce the number of bits per parameter from 16 `bits` to approximately 0.34 `bits` which is a 47x improvement in communication and storage costs. We now discuss two practical encoding schemes to realize these savings.

**Optimal Compression via Golomb Coding.** For maximal compression in communication and storage, Golomb coding is effective. This near-entropy method suits geometrically distributed data, like distances in sparse task vectors (Strom, 2015; Sattler et al., 2019a). Using Golomb coding (Golomb, 1966), we communicate the locations of non-zero elements with an additional bit indicating each element's sign, achieving near-optimal compression. This approach needs a total of $-((1-k)\log_2(1-k) + k\log_2(\frac{k}{2})) * d + 16$ `bits` for storage, and its average bits per position $\bar{\mathtt{b}}_{pos}$, the calculation of which is detailed in the footnote below[2]. Unless otherwise specified, storage costs reported in our experiments assume Golomb coding.

**Efficient Computation and Communication via Two Binary Vectors.** Alternatively, for scenarios prioritizing computational efficiency, ComPEFT compressed task vector $\tilde{\tau}_{\mathtt{t}}$ can be represented using two binary masks, one signifying positive values and the other signifying negative values. Formally, we need to communicate $\tilde{\tau}_{\mathtt{t}}^+ = (\tilde{\tau}_{\mathtt{t}} == +1) \in \mathbb{R}^d$ and $\tilde{\tau}_{\mathtt{t}}^- = (\tilde{\tau}_{\mathtt{t}} == -1) \in \mathbb{R}^d$, and the scalar constant $\alpha * \sigma(\tau_{\mathtt{t}})$. Each binary mask needs `1 bit/parameter`, resulting in $2 * \mathtt{d} + 16$ `bits` for communicating the update. Note that this requires strictly more storage than the Golomb-based encoding described above because $-((1-k)\log_2(1-k) + k\log_2(\frac{k}{2})) < 2$. However, sparse ternary vectors allow for efficient matrix operations. For example, to efficiently compute the distance between $\tilde{\tau}_{\mathtt{t}_1}$ and $\tilde{\tau}_{\mathtt{t}_2}$, we can do an `XOR` ($\oplus$) followed by a `POPCNT` for each group of 64 parameters (i.e. two machine instructions on a 64-bit architecture) twice, once for the positive and once for the negative masks. The dot product can also be calculated by using bitwise `AND` operations to calculate positive contributions (both vectors have $+1$ or $-1$) and negative contributions (one vector has $+1$, the other $-1$). The final dot product is the difference between the sum of these contributions. Similarly, other operations such as addition can also be made faster, which could reduce the time when merging models. Thus, ComPEFT offers flexibility to use Golomb coding for optimal storage, and binary vectors for efficient computation.

## 3 Main Results

### 3.1 Compressing QLoRA Trained on LLaMA Models

**Experimental Setup.** We first explore the utility of ComPEFT in the setting of training QLoRA adapters (Dettmers et al., 2023) for the LLaMA models (Touvron et al., 2023a) with 7B, 13B, 33B, and 65B parameters. We follow the experimental setting from the QLoRA paper (Dettmers et al., 2023) and experiment with 8 recent instruction-following datasets that are diverse in terms of languages and dataset sizes. This collection includes datasets generated by language models (Alpaca (Taori et al., 2023), self-instruct (Wang et al., 2022), and unnatural-instructions (Honovich et al., 2022)), a multitask dataset (FLAN-v2 (Chung et al., 2022a)), two datasets created via human annotation and feedback (OASST1 (Köpf et al., 2023) and HH-RLHF (Bai et al., 2022)), and two hybrid datasets (Chip2 (LAION, 2023) and Longform (Köksal et al., 2023)). For each of these datasets, we reuse the checkpoints released with the QLoRA paper[3] to perform compression using Algorithm 1 and then evaluate the 5-shot performance of the compressed QLoRA module

---

[2]Similar to Strom (2015); Sattler et al. (2019a;b), the average bits per position $\bar{\mathtt{b}}_{pos}$ is calculated as follows: $\bar{\mathtt{b}}_{pos} = \mathbf{b}^* + \frac{1}{1-(1-p)^{2^{\mathbf{b}^*}}}$, with $\mathbf{b}^* = 1 + \lfloor \log_2(\frac{\log(\phi-1)}{\log(1-p)}) \rfloor$ and $\phi = \frac{\sqrt{5}+1}{2}$ being the golden ratio.

[3]https://huggingface.co/timdettmers?search_models=qlora

Table 1: **Performance improvement from `ComPEFT` increases as models get bigger.** We present the performance (storage size in GB) on the MMLU Test for the original and compressed QLoRA models. For `LLaMA − 65B`, `ComPEFT` leads to a `4.16%` improvement while being `26x` smaller.

| Model Size (→) | 7B | | 13B | | 33B | | 65B | |
|---|---|---|---|---|---|---|---|---|
| Dataset (↓) | Original | ComPEFT | Original | ComPEFT | Original | ComPEFT | Original | ComPEFT |
| Self-Instruct | $36.45_{(0.3)}$ | $\mathbf{37.72}_{(0.03)}$ | $36.20_{(0.47)}$ | $\mathbf{45.15}_{(0.01)}$ | $50.98_{(0.91)}$ | $\mathbf{57.02}_{(0.02)}$ | $55.34_{(1.49)}$ | $\mathbf{63.43}_{(0.03)}$ |
| Longform | $34.37_{(0.3)}$ | $\mathbf{35.48}_{(0.02)}$ | $45.70_{(0.47)}$ | $\mathbf{46.80}_{(0.02)}$ | $54.60_{(0.91)}$ | $\mathbf{57.07}_{(0.07)}$ | $59.49_{(1.49)}$ | $\mathbf{62.95}_{(0.05)}$ |
| Chip2 | $34.88_{(0.3)}$ | $\mathbf{36.11}_{(0.02)}$ | $44.19_{(0.47)}$ | $\mathbf{45.06}_{(0.03)}$ | $51.72_{(0.91)}$ | $\mathbf{56.43}_{(0.03)}$ | $57.30_{(1.49)}$ | $\mathbf{63.32}_{(0.05)}$ |
| HH-RLHF | $\mathbf{35.52}_{(0.3)}$ | $35.30_{(0.01)}$ | $44.66_{(0.47)}$ | $\mathbf{44.99}_{(0.01)}$ | $53.41_{(0.91)}$ | $\mathbf{56.97}_{(0.07)}$ | $58.79_{(1.49)}$ | $\mathbf{63.42}_{(0.05)}$ |
| Unnatural Instruct | $\mathbf{42.14}_{(0.3)}$ | $41.82_{(0.02)}$ | $\mathbf{48.98}_{(0.47)}$ | $48.42_{(0.03)}$ | $56.65_{(0.91)}$ | $\mathbf{58.07}_{(0.09)}$ | $59.50_{(1.49)}$ | $\mathbf{63.30}_{(0.03)}$ |
| Guanaco (OASST1) | $35.02_{(0.3)}$ | $\mathbf{36.31}_{(0.01)}$ | $\mathbf{48.50}_{(0.47)}$ | $47.10_{(0.03)}$ | $55.51_{(0.91)}$ | $\mathbf{57.55}_{(0.05)}$ | $60.67_{(1.49)}$ | $\mathbf{63.25}_{(0.09)}$ |
| Alpaca | $\mathbf{40.72}_{(0.3)}$ | $39.95_{(0.02)}$ | $\mathbf{49.53}_{(0.47)}$ | $48.41_{(0.03)}$ | $53.66_{(0.91)}$ | $\mathbf{57.68}_{(0.05)}$ | $60.51_{(1.49)}$ | $\mathbf{63.28}_{(0.05)}$ |
| FLAN v2 | $43.97_{(0.3)}$ | $\mathbf{44.70}_{(0.02)}$ | $50.45_{(0.47)}$ | $\mathbf{50.76}_{(0.03)}$ | $56.67_{(0.91)}$ | $\mathbf{60.01}_{(0.07)}$ | $62.72_{(1.49)}$ | $\mathbf{64.61}_{(0.11)}$ |
| Average | $37.88_{(0.3)}$ | $38.42_{(0.0188)}$ | $46.03_{(0.47)}$ | $47.09_{(0.024)}$ | $54.15_{(0.91)}$ | $57.60_{(0.056)}$ | $59.29_{(1.49)}$ | $63.45_{(0.058)}$ |
| Increase/Comp. | − | +0.54 / 16x | − | +1.06 / 20x | − | +3.44 / 16x | − | +4.16 / 26x |

Table 2: **LLaMA2-70B Results.** Mirroring our main findings, `ComPEFT` improves average performance (by `1.69%`) on `LLaMA2 − 70B`, notably by `4.82%` on Self-Instruct.

| Dataset (↓) | Original | ComPEFT |
|---|---|---|
| **Alpaca** | 67.13 | 67.56 (+0.43) |
| **Chip2** | 65.18 | 67.00 (+1.82) |
| **Longform** | 67.63 | 68.50 (+0.86) |
| **Guanaco** | 66.89 | 67.39 (+0.5) |
| **Self-Instruct** | 62.36 | 67.18 (+4.82) |
| **Average** | 65.84 | 67.53 (+1.69) |

Table 3: `ComPEFT` **can compress smaller model with minimal performance loss.** Test set performance$_{(\text{Storage Size in MB})}$ averaged over seven GLUE tasks when compressing $(IA)^3$ and LoRA modules on different base models.

| PEFT (↓) | Method (↓) | T5-Base | T5-Large | T0-3B |
|---|---|---|---|---|
| $(IA)^3$ | Original | $81.3_{(0.25)}$ | $86.2_{(0.66)}$ | $89.3_{(1.03)}$ |
| | ComPEFT | $80.0_{(0.01)}$ | $85.9_{(0.04)}$ | $88.4_{(0.06)}$ |
| | Improvement | -1.3 / 25x | -0.3 / 16x | -0.9 / 17x |
| LoRA | Original | $79.2_{(6.19)}$ | $84.5_{(16.50)}$ | $89.5_{(33.75)}$ |
| | ComPEFT | $78.1_{(0.35)}$ | $84.6_{(1.37)}$ | $89.5_{(2.60)}$ |
| | Improvement | -1.1 / 17x | +0.1 / 12x | 0.0 / 13x |

on the MMLU benchmark (Hendrycks et al., 2020). To ensure the generalizability of `ComPEFT`, we extend our evaluation to `LLaMA2 − 70B` model. In all experiments, we sweep both $\alpha$ and $k$ in the following ranges, $k \in \{5, 10, 20, 30, 50\}$ and $\alpha \in \{0.5, 1, 2, 3, 4, 5, 6, 8, 10\}$ and report the storage size based on the entropy of `ComPEFT` as specified in §2.2. We find that at any given value of $k$, you can achieve good performance (see § 4.2). We used a single 48GB NVIDIA A6000 GPU for these experiments.

**Outcomes.** In Table 1, we provide results for all the task and model size combinations, comparing the performance of the `ComPEFT` checkpoints and the original QLoRA checkpoints along with (in subscripts) the storage size in GB assuming 16-bit precision for uncompressed models and Golomb code-based compression. We find that on `28` of `32` experimental configurations `ComPEFT` improves upon the performance of the original QLoRA models while compressing the LoRA module between `10x − 50x` in terms of storage costs. `ComPEFT` leads to an improvement of `0.54%`, `1.06%`, `3.44%`, and `4.16%` on MMLU for the `LLaMA 7B`, `13B`, `33B`, and `65B` parameter models, respectively. In Table 2, we observe similar compression and improvements for `LLaMA2 − 70B` model. To sum, `ComPEFT` provides better results while also reducing the QLoRA module size. For example, on the `65B LLaMA` base model it reduces the storage size from `1.5GB` to `110MB` while improving the MMLU performance by a large margin of `4.16%`.

**Discussion.** A few important conclusions about `ComPEFT` can be derived from these results: (1) `ComPEFT` can compress all QLoRA models by a factor of at least `10x`. (2) Larger base models allow for more compressible LoRA modules. We get a compression factor of approximately `16x`, `20x`, `16x`, and `26x` for 7B, 13B, 33B, and `65B` parameter models respectively. (3) A similar trend is found in performance – the performance gap between the original and the compressed LoRA module increases with model size from `0.54%` for the `7B` model to `4.16%` for the `65B` model. If this scaling law continues, it means that the utility of methods like `ComPEFT` will increase as models become larger and/or their zero-shot performance improves.

Table 4: **ComPEFT extends compression benefits to fully fine-tuned residuals.** Average test set performance (and storage size in GB) over 7 GLUE tasks for original and ComPEFT-compressed fully fine-tuned model's task vectors. Across various model architectures and sizes, ComPEFT compresses models by 12x − 19x with near-lossless performance, and even slight improvements for some models.

| Model ($\downarrow$) | Original | ComPEFT | Improvement |
|---|---|---|---|
| BERT − Base | $87.2_{(0.21)}$ | $86.8_{(0.011)}$ | -0.4 / 19x |
| BERT − Large | $86.3_{(0.64)}$ | $86.1_{(0.036)}$ | -0.2 / 18x |
| RoBERTa − Base | $85.5_{(0.24)}$ | $83.3_{(0.013)}$ | -2.2 / 18x |
| RoBERTa − Large | $88.6_{(0.68)}$ | $89.2_{(0.052)}$ | +0.6 / 13x |
| T5v.1 − Base | $74.1_{(0.47)}$ | $75.8_{(0.032)}$ | +1.7 / 15x |
| T5v.1 − Large | $84.0_{(1.5)}$ | $82.2_{(0.11)}$ | -1.8 / 14x |
| T5 − Base | $82.8_{(0.43)}$ | $78.1_{(0.032)}$ | -4.7 / 13x |
| T5 − Large | $85.2_{(1.41)}$ | $84.7_{(0.12)}$ | -0.5 / 12x |

Table 5: **ComPEFT enables order-of-magnitude reduction in model transmission and loading latency.** We report the wall clock time (mean and standard deviation) for two practical scenarios: downloading LLaMA model checkpoints (7B-65B) from a simulated internet server to a local machine, and transferring checkpoints from CPU to GPU memory, comparing original and ComPEFT-compressed versions. ComPEFT reduces download times by up to 32x and CPU-to-GPU loading times by up to 25x, highlighting the practical advantages in deployment and serving efficiency.

| Model ($\downarrow$) | Internet $\rightarrow$ Local (seconds) | | CPU $\rightarrow$ GPU (milliseconds) | |
|---|---|---|---|---|
| | Original | ComPEFT | Original | ComPEFT |
| LLaMA − 7B | $11.21_{2.44}$ | $1.16_{0.04}$ | $134.28_{4.76}$ | $11.23_{5.22}$ |
| LLaMA − 13B | $16.85_{3.83}$ | $1.75_{0.30}$ | $186.60_{5.42}$ | $23.09_{0.78}$ |
| LLaMA − 33B | $32.31_{6.76}$ | $2.46_{0.12}$ | $307.29_{55.59}$ | $18.00_{4.34}$ |
| LLaMA − 65B | $83.17_{9.14}$ | $2.59_{0.14}$ | $475.26_{66.51}$ | $18.60_{5.67}$ |

## 3.2 Compressing Other PEFT Updates

The finding that scaling the base model makes the PEFT modules more compressible and more performant brings up the question as to whether ComPEFT is still effective at smaller scales. We perform experiments on two widely used PEFT methods, $(IA)^3$ (Liu et al., 2022) and LoRA (Hu et al., 2021), with three models, T5-Base and T5-Large (Raffel et al., 2020a), and T0-3B (Sanh et al., 2021b). Specifically, we compress $(IA)^3$ and LoRA modules trained on 7 classification tasks from the GLUE benchmark (Wang et al., 2018a) belonging to three categories: Natural Language Inference (MNLI (Williams et al., 2018), RTE (Bentivogli et al., 2009), QNLI (Rajpurkar et al., 2016), WNLI (Levesque et al., 2012a)), Sentiment Analysis (SST2 (Socher et al., 2013)), and Paraphrase Detection (MRPC (Dolan & Brockett, 2005), QQP (Wang et al., 2018a)). For hyperparameter selection $(\alpha, k)$, we use the same grid and validation procedure as described in §3.1.

**Outcomes.** In Table 3, we present the average performance on the 7 aforementioned GLUE tasks (per-dataset results are provided in Appendix C.6) along with the average checkpoint size in MB (in subscripts) for three base models with both $(IA)^3$ and LoRA adapters. We find that even with smaller base models, ComPEFT compress the PEFT modules by a factor of 12x − 25x with minimal to no loss in performance. These results demonstrate that even at smaller scales ComPEFT can lead to substantial compression. Additionally, we performed some experiments with BERT (Devlin et al., 2018), RoBERTa (Liu et al., 2019a), and T5v1.1 (Raffel et al., 2020a) models that are not multitask-trained and/or have weak zero-shot performance (i.e. they generally require additional finetuning to perform well on any downstream tasks). We present the results for these models in Appendix C.7, where we observe that compression works well for LoRA with minimal performance loss. However, for $(IA)^3$ we observe significant performance drops which suggest that zero-shot performance may be important to enable ComPEFT compression of $(IA)^3$-based models.

## 3.3 Compressing Fully Fine-tuned Models

**Experimental Setup.** To assess the broader applicability of ComPEFT, we investigate its effectiveness beyond PEFT modules and explore its ability to compress task vectors produced by full-model fine-tuning. We adopt the experimental setting from § 3.2 to fine-tune the 7 classification tasks from the GLUE benchmark using full fine-tuning, and then compress the resulting fully fine-tuned task vectors using ComPEFT. We evaluate our method on four different model architectures – BERT (Devlin et al., 2018), RoBERTa (Liu et al., 2019a), T5-v1.1 (Raffel et al., 2020a), and T5 (Raffel et al., 2020a) – across two model sizes (Base and Large) for each architecture.

**Outcomes.** Table 4 presents the average test set performance over the 7 GLUE tasks for both original and ComPEFT-compressed fully fine-tuned modelsWe observe that ComPEFT effectively compresses fully fine-tuned models, achieving 12x − 19x compression ratios with minimal performance degradation. Notably, for

T5v1.1-base and RoBERTa-large models, `ComPEFT` even leads to performance improvements of `1.7%` and `0.6%` respectively, while simultaneously reducing model size by `15x` and `13x`.

**Discussion.** These results demonstrate that `ComPEFT` is not limited to compressing parameter-efficient modules but can also be effectively applied to compress fully fine-tuned models. This broader applicability expands the potential use cases of `ComPEFT` and highlights its versatility as a general model compression technique. The observed performance improvements in some cases, even with full fine-tuning compression, further suggest that `ComPEFT` may act as a regularizer, potentially improving generalization.

### 3.4 Reduced Transmission Cost and Loading Latency

**Experimental Setup.** A key practical advantage of model compression is reduced storage and transmission costs. To quantify these benefits for `ComPEFT`, we measure the wall clock time for two representative scenarios: (1) downloading a model checkpoint from a simulated internet server to a local machine, and (2) loading a model checkpoint from CPU memory to GPU memory. We perform these measurements for both original QLoRA checkpoints and their `ComPEFT`-compressed counterparts for LLaMA models of sizes `7B`, `13B`, `33B`, and `65B`For each scenario and model configuration, we repeat the measurement `10` times and report the mean and standard deviation of the wall clock times.

**Outcomes.** Table 5 presents the results for transmission latency and loading timeAs expected, `ComPEFT`-compressed checkpoints exhibit significantly reduced transmission and loading times compared to the original checkpoints across all model sizes and both scenariosFor example, downloading the `LLaMA − 65B ComPEFT` checkpoint from the internet is approximately `32x` faster than downloading the original checkpoint. Similarly, loading the `ComPEFT`-compressed `LLaMA − 65B` checkpoint from CPU to GPU is about `25x` faster.

**Discussion.** These measurements showcase `ComPEFT`'s substantial practical value beyond their size. The order-of-magnitude speedups in download and loading/offloading times significantly accelerate model deployment and improve multi-expert serving efficiency, especially in dynamic or resource-limited settings`ComPEFT` thus offers key real-world advantages for efficient model utilization.

### 3.5 `ComPEFT` vs. Other PEFT Methods

**Experimental Setup.** Next, we compare the $(IA)^3$ and LoRA checkpoints compressed by `ComPEFT` with various other PEFT methods to determine whether `ComPEFT` produces Pareto-optimal parameter-efficient fine-tuning in terms of Storage Size and Performance. For this, we use the T0-3B (Sanh et al., 2021b) model and train a wide range of PEFT methods on the `11` held-out datasets from Sanh et al. (2021b) – specifically, sentence completion (COPA (Roemmele et al., 2011), H-SWAG (Zellers et al., 2019), and Story Cloze (Sharma et al., 2018) datasets), natural language inference (three splits of ANLI (Nie et al., 2019), CB (Marneffe et al., 2019), and RTE (Dagan et al., 2005)), coreference resolution (WSC (Levesque et al., 2012b) and Winogrande (Sakaguchi et al., 2020)), and word sense disambiguation (WiC (Pilehvar & Camacho-Collados, 2019)). For each task, from the training set, we select `200` example for the validation set and then use the first template from Prompt Source (Bach et al., 2022) both dur-

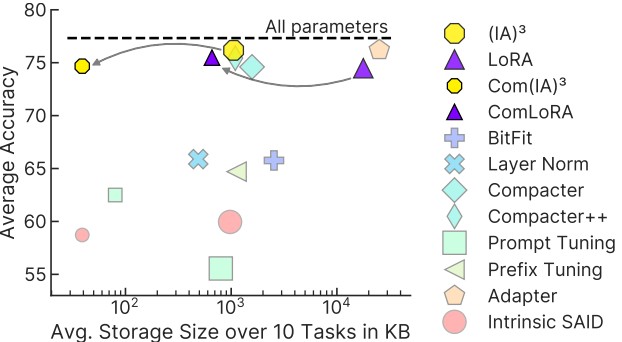

Figure 3: `ComPEFT` **are Pareto-optimal.** Performance vs storage size for multiple PEFT methods averaged over `11` tasks. A PEFT method is Pareto-optimal if it attains better performance (higher on the y-axis) than all methods that use less storage space (to the left on the x-axis). In particular, `Com(IA)³` performance is comparable to PEFT methods that require `1000×` more storage space.

ing training and evaluation. We perform experiments with `10` different PEFT methods from Liu et al. (2022) – LoRA (Hu et al., 2021), $(IA)^3$ (Liu et al., 2022), BitFit (Zaken et al., 2021), LayerNorm, Adapters (Houlsby

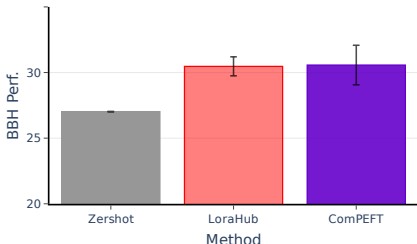

Figure 4: `ComPEFT` **facilitates compositional generalization.** Average performance of LoraHub and `ComPEFT` for compositional generalization on BBH.

Table 6: `ComPEFT` **compressed checkpoints lead to better merged models.** Average test set results on 7 GLUE tasks when employing different merging methods on the uncompressed checkpoints and compressed `ComPEFT` checkpoints.

| Method ($\downarrow$) | T5-Base | | T5-Large | | T0-3B | |
|---|---|---|---|---|---|---|
| | $(IA)^3$ | LoRA | $(IA)^3$ | LoRA | $(IA)^3$ | LoRA |
| Averaging | 53.7 | 49.3 | 55.4 | 50.2 | 74.5 | 73.1 |
| Task Arithmetic (TA) | **60.0** | 52.8 | **62.7** | 61.6 | 77.8 | 75.0 |
| ComPEFT + TA | 59.7 | **53.9** | 61.9 | **64.9** | **80.0** | **75.6** |
| TIES-Merging | 55.5 | 49.2 | **61.3** | 57.3 | 71.7 | 73.4 |
| ComPEFT + TIES | **55.6** | **49.2** | 60.4 | **61.4** | **76.2** | **75.8** |

et al., 2019), Compacter and Compactor++ (Karimi Mahabadi et al., 2021), Prompt Tuning (Lester et al., 2021), Prefix Tuning (Li & Liang, 2021), and Intrinsic SAID (Aghajanyan et al., 2020).

**Outcomes.** In Figure 3, we plot the average performance over the 11 tasks and the checkpoint sizes in KB for 10 PEFT Method and when using `ComPEFT` on LoRA and $(IA)^3$ checkpoints, i.e. `ComLoRA` and `Com(IA)`$^3$. We find that `ComPEFT` reduces the storage size for both LoRA and $(IA)^3$ by more than an order of magnitude with minimal reduction in performance. From this plot, `ComPEFT` is Pareto-optimal, i.e. for any given storage budget, `ComPEFT` outperforms all other PEFT methods. Notably, `Com(IA)`$^3$ exhibits only a minor performance degradation compared to full-model fine-tuning while being one of the most space-efficient PEFT methods. Lastly, we note that for `ComPEFT` you can trade-off performance for storage cost by varying the density $k$ to obtain models of different sizes. Hence, `Com(IA)`$^3$ and `ComLoRA` could be made even more space efficient.

**Discussion.** Figure 3 positions `ComPEFT` as a highly competitive PEFT technique, not just a compression methodWhile methods like Prompt Tuning and Prefix Tuning offer parameter efficiency, `ComPEFT`, especially `Com(IA)`$^3$, achieves a significantly better balance of performance and size. Traditional full fine-tuning, while offering slightly higher peak performance, is orders of magnitude larger in sizeThis Pareto-optimality highlights the practical advantages of `ComPEFT` in resource-constrained scenarios. Furthermore, while we focus on Pareto optimality against other PEFT methods here, `ComPEFT` also provides a strong compression baseline for any PEFT technique; applying `ComPEFT` to other PEFT outputs could further enhance their storage efficiency.

### 3.6 Cross-Task Generalization via Dynamic LoRA Module Composition

**Experimental Setup.** As highlighted in §1, a key motivation is to enable efficient serving of numerous experts, particularly in scenarios requiring dynamic adaptation to novel tasks. Cross-task generalization, using compositional methods like LoraHub which dynamically select, load, and compose expert modules, exemplifies such a scenario where communication bottlenecks are critical. Therefore, to assess `ComPEFT`'s ability to facilitate efficient expert module serving in this demanding downstream application, we examine its impact on the composability of the resulting PEFT modules for cross-task generalization. Given a set of expert models and an unseen downstream task with few training examples, the goal is to combine a subset of these expert modules to attain a model that performs well on the unseen task.

For this, we follow the LoraHub (Huang et al., 2023) method and their experimental setting. We use the Flan-T5-large (Chung et al., 2022b) model as it exhibits strong zero-shot and few-shot capabilities. We consider nearly 200 distinct (tasks, instruction) pairs that were utilized to train the Flan-T5 model and use the LoRA modules trained on these tasks as expert[4]. Following Huang et al. (2023), when learning a new unseen task, we randomly select N LoRA modules denoted by $\{L_i = (A_i, B_i)\}_{i=1}^N$ and compose them as

$$L_m = A_m B_m = \left(\sum_{i=1}^N w_i A_i\right)\left(\sum_{i=1}^N w_i B_i\right), \tag{1}$$

---

[4]hf.co/models?search=lorahub

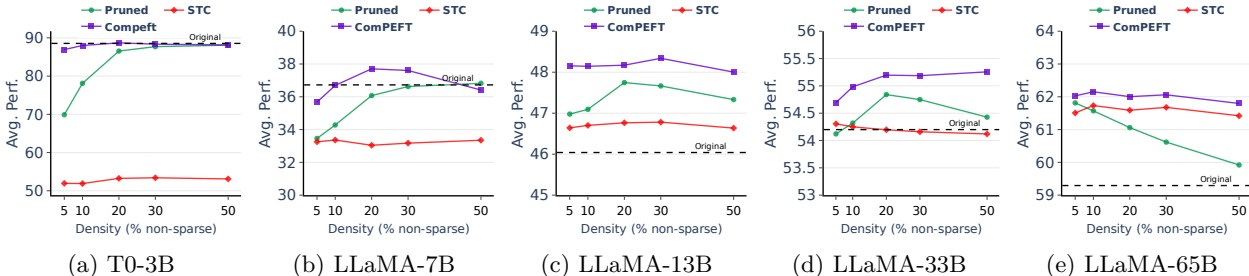

(a) T0-3B     (b) LLaMA-7B     (c) LLaMA-13B     (d) LLaMA-33B     (e) LLaMA-65B

Figure 5: `ComPEFT` **outperforms STC and all method steps are crucial.** Average performance as density `k` of the compressed checkpoint increases. We show results for compressing LoRA modules trained over different models with sizes ranging from `3B − 65B` and compare them with baselines and ablate method components.

where $A_m$, $B_m$ are the matrics of the composed LoRA module and $w_i$ are parameters that are learned on the few-shot examples from the unseen tasks using the gradient-free Shiwa optimizer (Liu et al., 2020). Following LoraHub, we use `N = 20` and treat the `27` diverse tasks from the Big-Bench Hard (BBH) benchmark (Suzgun et al., 2022) as our unseen evaluation tasks. All the tasks are multiple-choice questions and we employ Exact Match (EM) as our evaluation metric. Error bars in Figure 4 represent standard deviation across the BBH tasks performance.

**Outcomes.** In Figure 4, we report the average performance and standard deviation (over 5 seeds) when using the LoraHub method on the original checkpoints and the `ComPEFT`-compressed checkpoints. We find that the `ComPEFT`-compressed checkpoints exhibit similar compositional abilities as the original uncompressed checkpoints. This is a crucial finding: even with extreme compression, the modules retain the necessary properties for effective cross-task composition. Hence, `ComPEFT` checkpoints can be communicated quickly over high latency networks for dynamic module swapping, while maintaining their compositional abilities.

**Discussion.** The preservation of compositional generalization performance after `ComPEFT` compression is a significant result. It directly addresses the practical challenge of serving numerous expert modules for complex tasksBy drastically reducing module size, `ComPEFT` makes dynamic module composition via methods like LoraHub far more efficient and scalable, enabling faster download times and reduced memory footprint during run-time module swappingThis experiment validates that `ComPEFT` is not only a compression technique but also a facilitator for advanced applications requiring efficient expert module management and communication.

### 3.7 Merging Compressed PEFT Modules

**Experimental Setup.** Next, we examine the effectiveness of `ComPEFT` when merging models (Choshen et al., 2022; Matena & Raffel, 2021) by comparing the merging of compressed or uncompressed models. We follow the experimental setting (including base models, PEFT methods, and datasets) from the previous section and merge the `7` GLUE tasks to produce a multitask model. We then report the average performance of the merged across all tasks. We use two methods to merge task vectors, namely, Task Arithmetic (Ilharco et al., 2023) and TIES-Merging (Yadav et al., 2023). We used the code from the original authors for both merging methods.

**Outcomes.** As demonstrated in Table 6, in `9` out of `12` scenarios, the `ComPEFT` checkpoints lead to better merged models compared to the original checkpoints, with the notable exception of $(IA)^3$ on T5 models. Notably, in stronger models like T0-3B, `ComPEFT`-compressed checkpoints not only reduce the size by approximately `15x` but also improve the merged model's performance by `2.4%` on average. One possible explanation is that `ComPEFT` acts as a regularizer, removing less important parameter updates and potentially leading to smoother, more generalizable loss landscape that merge more effectively. This shows `ComPEFT`'s efficacy in both minimizing storage and communication overheads and improving the model merging performance.

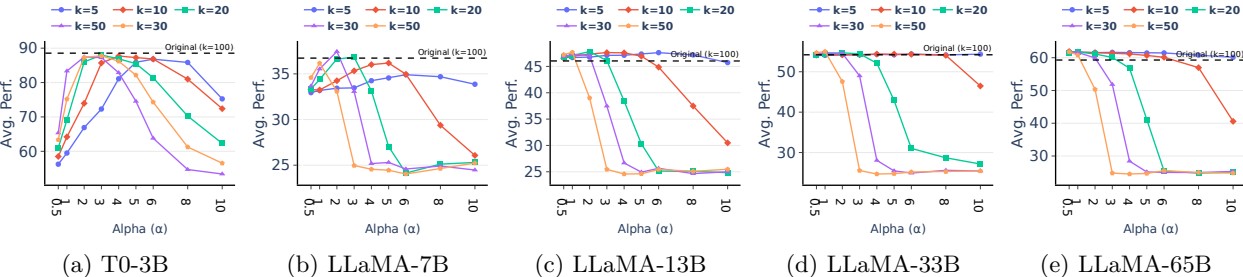

| (a) T0-3B | (b) LLaMA-7B | (c) LLaMA-13B | (d) LLaMA-33B | (e) LLaMA-65B |

Figure 6: **Larger models do not require explicit tuning of $\alpha$.** Performance vs $\alpha$ for various denisty levels for `ComPEFT`.

## 4    Additional Results and Analysis

### 4.1    Ablation of `ComPEFT` Components

**Experimental Setup.** To understand the contribution of the individual steps of `ComPEFT`, we now perform a brief ablation study. In `ComPEFT` there are two main steps: (1) Sparsifying the direction vectors, and (2) Quantizing the magnitudes to a scalar with scaling factor $\alpha$. Hence, we compare with two ablated versions: *Pruned* (only sparsification, magnitudes reset to zero, no quantization, no scaling), and Sparse Ternary Compression ($STC$) (Sattler et al., 2019a) (ternary quantization with mean magnitude scaling, no tuned $\alpha$). We also include the uncompressed *original* model as a baseline. We provide these ablations for the experimental settings from § 3.1 and 3.5 where the model sizes range from 3B − 65B.

**Outcomes.** In Figure 5, we plot the average validation set performance over tasks as a function of the density ($k$) of the compressed model. From these results, we make a few observations: (1) `ComPEFT` almost always performs better than both STC and the Pruned version for all model sizes and sparsity levels. (2) `ComPEFT` almost always performs better than or similar to the original model's performance for all sparsity levels. In contrast, for smaller model sizes of 3B and 7B, STC's performance is much worse than the original models. This highlights the importance of the scaling $\alpha$ as proposed in `ComPEFT`, which allows us to recover the performance lost due to pruning and ternary compression without computationally expensive retraining. (3) At low density, the performance of *Pruned* is much worse than `ComPEFT` and this gap reduces as the density increases. However, note that the size of `ComPEFT` is much smaller than the *Pruned* baseline due to ternarization. (4) At larger base model sizes ($\geq$ 13B), all the methods at all density levels perform similarly to or better than the original LoRA checkpoint, suggesting increased robustness to compression choices at scale.

**Discussion.** The ablation study clearly demonstrates the contribution of each component. The superior performance of `ComPEFT` over the *Pruned* variant underscores the importance of ternary quantization and scalar scaling in maintaining performance after sparsification. The advantage over STC highlights the benefit of tuning the scaling factor $\alpha$ rather than using a fixed magnitude scaling like mean magnitude. These results validate our design choices and show the utility of sparsification, ternarization, and tuned scalar scaling.

### 4.2    Effect of Sparsity and Scaling on `ComPEFT`

**Experimental Setup.** For `ComPEFT`, we analyze the effect of different levels of sparsity and the scaling value $\alpha$ on the performance of the compressed checkpoints. We present this analysis for T0 − 3B and LLaMA as the base models; the experimental settings are similar to § 3.1 and 3.5 where the model sizes range from 3B − 65B. We provide results for different values of the density $k$ (sparsity = 100 − k), specifically, the values $k \in \{5, 10, 20, 30, 50\}$ and different values of $\alpha \in \{0.5, 1, 2, 3, 4, 5, 6, 8, 10\}$.

**Outcomes.** In Figure 6, we plot the average validation set performance across all tasks with respect to the scaling coefficient $\alpha$. We make the following observations; (1) For smaller base-model sizes (3B and 7B) and across density values, we find a similar trend – as the value of $\alpha$ increases, the average validation performance first increases and then drops. (2) As the value of $k$ increases, the optimal value of $\alpha$ tends to becomes smaller. For example, for the T0-3B base model, the optimal value $\alpha$ for $k = 50$ is between 2 − 3 while for $k = 5$ the optimal $\alpha$ is in the range 5 − 8. (3) For bigger base-models ($\geq$ 13B) and low density ($k \in \{5, 10, 20\}$) the

variation in performance as $\alpha$ changes is smaller. (4) Lastly, as the base-model size increases, smaller values of $\alpha \in (0.5, 2)$ and a bigger range of values start to work better. Hence, for large models, the need for tuning $\alpha$ can be removed. For models with $\geq$ 13B parameters and high sparsity $k \leq$ 20, we recommend setting $\alpha = 1$.

**Discussion.** This analysis highlights the interplay between sparsity and scaling in ComPEFT. For smaller models, fine-tuning $\alpha$ is important to maximize performance at a given sparsity level. However, for larger models, the method becomes more robust, and a fixed scaling factor (like $\alpha = 1$) can be sufficient, especially at higher sparsity. This robustness for larger models simplifies the application of ComPEFT in practice, as it reduces the need for extensive hyperparameter search and makes the method more readily deployable for large language models where computational efficiency is paramount.

## 5 Related Work

**Paremeter Efficient Fine-Tuning.** Several parameter-efficient techniques (Lester et al., 2021; Li & Liang, 2021; Houlsby et al., 2019; Zaken et al., 2021) have emerged as efficient alternatives to full fine-tuning in the field of pre-trained language models (PLMs). These methods introduce a small number of additional parameters to PLMs (Raffel et al., 2020a; Touvron et al., 2023a) enabling efficient fine-tuning. LoRA (Hu et al., 2021) incorporates trainable low-rank matrices into transformer layers. In Contrast, $(IA)^3$ (Liu et al., 2022) learns a new set of parameters to rescale the model activations. Recently, QLoRA (Dettmers et al., 2023) proposed training LoRA modules over a 4-bit quantized base model to further save the memory.

**Network Pruning and Federated Learning.** Neural network pruning techniques have garnered attention for reducing computational costs (Cheng et al., 2017; Liang et al., 2021) by removing redundant parameters while preserving performance (Zhu & Gupta, 2018; Liu et al., 2019b; Frankle & Carbin, 2019; Gale et al., 2019; Xia et al., 2022). Among these, magnitude-based pruning (Han et al., 2015; Li et al., 2018; Lee et al., 2021) selects parameters based on magnitudes. Pruning is valuable in federated learning due to high communication costs over slow networks. Atomo (Wang et al., 2018b) minimizes gradient variance through unbiased sparsification, while QSGD (Alistarh et al., 2017) offers a communication-convergence trade-off by quantizing gradients. SignSGD (Bernstein et al., 2018) further converts gradients to binary sign vectors. TernGrad (Wen et al., 2017) and STC (Sattler et al., 2019a) combine sparsification and quantization.

**Model Merging and Compositional Generalization.** Various merging methods (Ortiz-Jiménez et al., 2023; Wortsman et al., 2022b;a; Ilharco et al., 2022; Ramé et al., 2022; Yu et al., 2023) aim to combine fine-tuned models for improved performance in various applications. Choshen et al. (2022) performs direct averaging of the model weights while Task Arithmetic (Ilharco et al., 2023) generates task vectors and performs arithmetic operations to create multitask checkpoints. Ortiz-Jiménez et al. (2023) offer theoretical insights into model merging by using the weight disentanglement property. TIES-Merging (Yadav et al., 2023) identifies the issue of parameter interference in model merging and tackles it by trimming low-magnitude parameters, resolving sign disagreements, and disjointly merging parameters with consistent signs. Ponti et al. (2023) performed CG by jointly learning adapters and a routing function to allocate skills to tasks, while Caccia et al. (2023) analyzes task routing for more efficient cross-task generalization. LoraHub (Huang et al., 2023) employs gradient-free optimization to retrieve and merge expert modules for unseen tasks while Muqeeth et al. (2024) focus on zero shot compositional generalization. Pfeiffer et al. (2023) provides an overview of PEFT methods, model merging, and compositional generalization methods.

## 6 Conclusion

Our PEFT compression method, `ComPEFT`, offers an effective solution to the latency challenges associated with retrieving expert models. By compressing fine-tuning residuals through sparsification and quantization, `ComPEFT` achieves high compression ratios and often enhances model performance across various NLP tasks and model sizes. Moreover, it preserves few-shot compositional generalization capabilities, facilitates efficient communication and computation, and demonstrates improved performance when merged with original models. This research contributes valuable insights into the realm of parameter-efficient fine-tuning, addressing both performance and latency concerns.

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

## A  Limitations

While `ComPEFT` demonstrates significant promise, it is important to consider several limitations of this work. Firstly, while average performance is strong, certain task types might exhibit reduced effectiveness or require specific hyperparameter tuning. Secondly, we observed some performance sensitivity with `ComPEFT` applied to $(IA)^3$ modules, especially on base models with weaker zero-shot capabilities, warranting further investigation into the interplay between base model properties, PEFT methods, and compression. From a practical standpoint, while hyperparameter tuning for the scaling factor $\alpha$ becomes less crucial for larger models, it remains relevant for smaller models, adding a hyperparameter selection step to their deployment. Future work could explore automated or adaptive methods to address this. Furthermore, a rigorous theoretical understanding of `ComPEFT` is still lacking. We do not have a definitive explanation for the observed performance improvements in some cases, nor why these improvements scale with model size, although noise reduction is a possible factor suggested by related works. A deeper understanding of how fine-tuning updates encode information and how `ComPEFT` interacts with this information is necessary for developing even more refined compression techniques. Finally, to fully unlock the potential wall-clock speedups promised by `ComPEFT`'s ternary vector representations, dedicated engineering effort is needed to develop custom Triton/CUDA kernels optimized for operations on sparse ternary data structures, as briefly discussed in our methodology section. These areas represent important avenues for future research, particularly in the context of efficiently serving and composing large numbers of expert PEFT modules for advanced applications.

## B  Implementation Details

### B.1  Training Details

In our research, we utilized the following models, BERT-base, BERT-Large, RoBERTa-base, RoBERTa-large, T5v1.1-base, T5v1.1-large, T5-base, T5-large, Flan-T5-large, T0-3B, LLaMA 7B, 13B, 33B, 65B models. The Flan-T5-Large and LLaMA models were not trained by us and were used by the authors of QLoRA (Dettmers et al., 2023) and LoraHub (Huang et al., 2023). For the experiments in §3.2 and §3.3 on the 7 GLUE (Wang

et al., 2018a) tasks, we trained the large datasets (mnli, qnli, sst2, qqp) for 1 epoch and the small datasets (rte, mrpc, wnli) for 10 epochs. Whereas for the experiment in §3.5, we followed most of the hyperparameter configuration from the $(IA)^3$ (Liu et al., 2022) paper and trained for 2500 steps with a batch size of 8. For each of the 11 datasets in §3.5, we selected 200 examples from the training set to be used as the validation set for best model selection as well as selecting the hyperparameters for ComPEFT. Across all experiments to obtain the trained models we selected different learning rates for each dataset and PEFT method. For training $(IA)^3$ models we selected the learning rate from $\{1e-2, 1e-3, 1e-4, 1e-5\}$, for LoRA from $\{5e-2, 5e-3, 5e-4, 5e-5\}$, and for full model finetuning from $\{5e-3, 5e-4, 5e-5, 5e-6\}$. During the training process, bfloat16 was adopted to curtail GPU memory expenditure. For the purpose of evaluation, models from the T5 and T0 families were evaluated using rank classification to select the correct label. In this method, the model's log probabilities for all potential label strings are ranked. The model's prediction is deemed accurate if the choice ranked highest aligns with the correct answer. It should be noted that rank classification evaluation can accommodate both classification tasks and multiple-choice tasks.

## B.2 Compute Resources Used and Runtimes

We executed all our experiments on Nvidia A6000 GPUs equipped with 48GB RAM. Training $(IA)^3$ and LoRA models on the T0-3B model for a single (§3.2, §3.5, and §3.7) task takes about 30 minutes to 4 hours depending on the dataset. For T5-Base and T5-Large models (§3.2, §3.7), based on dataset size, needed between 15 minutes and 2 hours per task. Experiments with QLoRA on LLaMA models were done using the original checkpoints from QLoRA paper (Dettmers et al., 2023) for all the 8 instruction tuning datasets and are supplied the authors of QLoRA here.[5] The ComPEFT compression experiments were efficient, with evaluations consuming between 10-30 seconds for the T5-Base, T5-Large, and T0-3B models. For LLaMA models, following QLoRA (Dettmers et al., 2023), the hyperparameter selection is done on a small held-out subset of MMLU (Hendrycks et al., 2020) benchmark and takes about 8 minutes, 14 minutes, 28 minutes, and 49 minutes for LLaMA 7B, 13B, 33B, and 65B models respectively.

## B.3 Employed Datasets and Associated Licences

We use the following datasets in the paper with the following licenses.
Apache License 2.0: Flan V2, Self-Instruct, Chip2
cc-by-nc-4.0: Alpaca
MIT License: Guanaco, Unatural Instructions, HH-RLHF, Longform
Not Found: GLUE

## B.4 Gradient Noise

Gradients are (almost) never 0 for any parameter, as all parameters somehow affect the result. Thus, we presume most updates in fine-tuning are not more than just noise, rather than learned updates. We compute the mean and standard deviation of the task vector of a LoRA model finetuned on LLaMa (Touvron et al., 2023a) base model and compare it with the base model. We find the mean of both the LoRA task vector and the base model is close to zero, however, the LoRA task vector has a small standard deviation of 0.0007 as compared to 0.0228 for the LLaMA base. This further confirms the hypothesis that most parameters are changed very little during fine-tuning.

## B.5 Why Use Multiplication Factor of Standard Deviation

This decision to use multiplication factors of std is based on a few observations: (1) the task vector parameters are typically normally distributed with almost zero means (see table below), implying that the pretrained parameters have not changed much on average and some specific parameters get huge updates while others just accumulate SGD noise. (2) This std of task vectors can differ a lot based on the model size and the dataset. (3) We only care about the top-k fraction of the parameters (say top-20%) that lie outside the first standard deviation ($> \sigma$), i.e., that has a magnitude greater than $\sigma$. Hence, given this different scale of

---

[5]https://huggingface.co/timdettmers?search_models=qlora

Table 7: Statistics of the distribution of the task vectors for different model sizes and datasets.

| Model ($\downarrow$) | Dataset ($\downarrow$) | $\text{TV}_{\text{mean}}$ | $\text{TV}_{\text{std}}$ | $\text{TV}_{\text{max}}$ | $\text{TV}_{\text{min}}$ |
|---|---|---|---|---|---|
| T0 $-$ 3B | **storycloze** | 1.61E-02 | 0.1347 | 31.9909 | -4.8564 |
| | **winogrande** | 1.61E-02 | 0.1357 | 32.3445 | -4.7301 |
| LLaMA $-$ 7B | **chip2** | -8.66E-07 | 0.0155 | 0.083 | -0.082 |
| | **longform** | -1.95E-06 | 0.0172 | 0.0859 | -0.085 |
| LLaMA $-$ 13B | **chip2** | 1.69E-06 | 0.0106 | 0.0762 | -0.0767 |
| | **longform** | -1.47E-06 | 0.0173 | 0.0767 | -0.0747 |
| LLaMA $-$ 33B | **chip2** | -2.39E-07 | 0.0095 | 0.0688 | -0.0703 |
| | **longform** | -8.80E-08 | 0.0075 | 0.0703 | -0.0708 |
| LLaMA $-$ 65B | **chip2** | -5.07E-07 | 0.0083 | 0.062 | -0.063 |
| | **longform** | -7.91E-08 | 0.0097 | 0.0635 | -0.064 |
| LLaMA2 $-$ 70B | **chip2** | 3.63E-07 | 0.0053 | 0.0396 | -0.0393 |
| | **longform** | -1.16E-07 | 0.0053 | 0.0427 | -0.043 |

top-k task vector parameters across different model sizes, tasks, etc., the standard deviation serves as a nice unifying scale that provides us with a constant set of values to try for $\alpha$, making this process simpler.

In Table 7, we provide the mean, standard deviation, maximum, and minimum values of the task vectors for models of different sizes and datasets. We observe that std, max, and min values change as the model size changes. For example, for the 3B model, the std is 0.13 while for the 70B model, the std is 0.009. Hence, we use $\alpha * \sigma$ as it allows us to try hyperparameters in the correct range. However, we agree that there might be other ways to go about selecting $\alpha$, for example, learning on a small dataset.

## C  Additional Results

### C.1  Comparision With Other Additional Pruning Methods

We performed additional experiments in a setting similar to Table-2, where we worked with the Llama-2 (Touvron et al., 2023a) 70B model and learned qLora (Dettmers et al., 2023) modules of rank 64. We then compressed these parameter updates using ComPEFT, STC, BitDelta (Liu et al., 2024a), and DAREx (Deng et al., 2024) methods. Note that the BitDelta method has two variants. The first variant does not perform any additional training for the scale parameter (referred to as "No Training"). In the "BitDelta (No Training)" setting, the scale parameter ($\alpha$) is set to the mean value of all the parameters in the task vector/delta weights. The "BitDelta (Training)" variant learns the scale parameter ($\alpha$) via SGD and hence is not directly comparable with our ComPEFT which requires no additional training. For the DAREx method we use the DAREx-q ($1/q_v$) variant, which uses labelled data to select the inverse scaling parameter ($q_v$) for each per-layer separately after pruning. We DAREx, we use sparsity levels of 95% and 99% as used in their paper. The results for the experiments are provided below along the average sizes of the compressed parameters across all the tasks.

From these results, we can clearly see that: (1) ComPEFT performs better than these baseline. (2) DAREx (p=0.95) and BitDelta(No Training) show slight performance loss compared to the original checkpoint while DAREx (p=0.99) results in a huge drop. This is in line with the results presented in their papers. (3) BitDelta (Training) performs similar to ComPEFT, however, this method learns the scalar ($\alpha$) which requires both forward and backward passes and hence more GPU memory. (4) Note that BitDelta (No Training) sets the scalar ($\alpha$) as the mean of all the values in the task vector. It is very similar to STC which also uses the mean value as the scalar. However, they have a critical difference which is that STC also performs sparsification before performing quantization. Hence, in BitDelta the values are (+a, -a) while in STC the values are like (+b,0,-b). We note that STC performs slightly better than BitDelta (No Training), we believe

Table 8: Comparing ComPEFT with other additional Pruning methods.

| Dataset | Original | ComPEFT | STC | BitDelta | | DAREx-$q_v$ | |
|---|---|---|---|---|---|---|---|
| | | | | No Training | Training | p=0.95 | p=0.99 |
| **alpaca-clean** | 67.13 | 67.56 | 66.57 | 66.27 | 67.43 | 65.85 | 39.57 |
| **chip2** | 65.18 | 67 | 64.54 | 64.31 | 67.31 | 63.94 | 50.18 |
| **longform** | 67.63 | 68.5 | 67.02 | 66.15 | 68.61 | 66.14 | 44.32 |
| **oasst1** | 66.89 | 67.39 | 66.15 | 65.38 | 67.11 | 65.48 | 45.82 |
| **self-instruct** | 62.36 | 67.18 | 61.94 | 61.52 | 66.82 | 61.97 | 49.39 |
| **Average** | 65.84 | 67.53 | 65.24 | 64.73 | 67.46 | 64.68 | 45.86 |
| **Size** | 1.58GB | 56MB | 56MB | 99MB | 99MB | 395MB | 79MB |

that this is due to the sparsification step which removes redundant parameters which add noise. Similar phenomenon is also observed in TIES-Merging (Yadav et al., 2023). Lastly, we also report the storage size for the compressed checkpoints where we use different methods to store them. We use golomb coding for ComPEFT/STC, bitmask for Bitdelta, and coo_sparse matrix for DAREx method. The results demonstrate that ComPEFT yields better performance/size trade-off compared to most of these other methods.

## C.2 Comparisons with Advanced PEFT Methods

Table 9: Comparison with other PEFT methods

| Dataset | LORA | ComLORA | DORA | ComDORA |
|---|---|---|---|---|
| **alpaca-clean** | 67.13 | 67.56 | 68.42 | 69.78 |
| **chip2** | 65.18 | 67 | 67.21 | 68.32 |
| **longform** | 67.63 | 68.5 | 69.36 | 68.92 |
| **oasst1** | 66.89 | 67.39 | 68.89 | 67.63 |
| **self-instruct** | 62.36 | 67.18 | 65.26 | 66.31 |
| **Average** | 65.84 | 67.53 | 67.83 | 68.19 |
| **Size** | 1.58GB | 56MB | 1.59GB | 57MB |

We conducted some additional experiments with some other PEFt methods like DoRA (Liu et al., 2024b). For the experimental setting in Table-2 with rank 64 LoRA on the Llama-2 70B model. We performed additional experiments with DoRA of rank 64 and then compressed them using ComPEFT and reported the results. We omitted VeRA (Kopiczko et al., 2023) methods as based on the DoRA paper VeRA typically performs worse than both LoRA and DoRA. Lastly, we omitted HiRA (Huang et al., 2025) as the method as due to its recency its code is not available. In Table 9 we present our results. Similar to our other finding, we observe that ComPEFT can also compress DoRA checkpoints to a great extent while preserving performance. Moreover, ComDoRA checkpoints slightly outperform ComLoRA's performance.

## C.3 Comparison of Compressed Lora With Lower Rank Lora Modules

We perform additional experiments to compare the compressed LoRA modules with lower rank lora module which inherently have smaller sizes as compression can be achieved on smaller rank. We opt for the experimental setting from Table-2 where we work with the Llama-2 70B model. We perform experiments with rank 32 and 8 the results of which are attached below along with storage sizes.

From the results in Table 10, we can see that: (1) at rank 32 there is a slight drop in performance compared to rank 64. We can compress the rank checkpoint as well by >25x (2) At rank 8, we see a significant drop in performance from 65.84 to 63.77. Moreover, ComPEFT can compress rank 8 lora as well by >25x. (3) for both rank 32 and 8, ComLoRA performs better than the original checkpoints. These results help us to

conclude that the observed benefits in compression and performance improvements stem from ComPEFT as opposed to the overparameterized LoRA adapter.

Table 10: Comparing ComPEFT with smaller rank LoRA modules.

| Dataset | Lora(r=64) | ComLora(r=64) | Lora(r=32) | ComLora(r=32) | Lora(r=8) | ComLora(r=8) |
|---|---|---|---|---|---|---|
| alpaca-clean | 67.13 | 67.56 | 66.98 | 67.24 | 64.82 | 65.27 |
| chip2 | 65.18 | 67 | 65.24 | 66.75 | 63.35 | 65.18 |
| longform | 67.63 | 68.5 | 67.14 | 68.12 | 65.16 | 66.74 |
| oasst1 | 66.89 | 67.39 | 65.42 | 66.92 | 64.21 | 65.56 |
| self-instruct | 62.36 | 67.18 | 62.68 | 67.48 | 61.32 | 65.81 |
| Average | 65.84 | 67.53 | 65.49 | 67.30 | 63.77 | 65.71 |
| Size | 1.58GB | 56MB | 790MB | 28MB | 197MB | 7MB |

## C.4 Validation Set Results

In Table 11 and 12, we provide the validation set results for our main compression experiments on `LLaMA`, `T5`, `T0` experiments from Section 3.1 and 3.2 respectively.

## C.5 Full Results for Compositional Generalization

In Table 13, we present the Zeroshot, ICL, LoraHub, and `ComPEFT` results for each of the BBH tasks.

## C.6 Individual Task Results

We present the task level validation and test set results along with model sizes of $(IA)^3$, LoRA, and full finetuning for T5-base (Table 20), T5-large (Table 21), T0-3B (Table 22).

## C.7 Compressing Model With Smaller Models with Bad ZeroShot Performance

We present the task level validation and test set results along with model sizes for $(IA)^3$, LoRA, and full finetuning for BERT-base (Table 14), BERT-large (Table 15), RoBERTa-base (Table 16), RoBERTa-large (Table 17), T5-v1.1-base (Table 18), and T5-v1.1-large (Table 19). These models are only trained using the pretraining objective and are not multitask-trained. Hence, these models have very bad zero/few-shot performance and always require explicit finetuning to perform well on any downstream tasks. We observe that for the LoRA method, the performance of `ComPEFT` is similar to the uncompressed full models while being smaller in size. This hints at the fact that the intrinsic dimensionality of the LoRA adaptation is much smaller compared to the number of parameters in the LoRA module. However, for $(IA)^3$ method, the performance drop is more, we believe that two reasons for this are: (1) The models are not good zero/few-shot models, and (2) $(IA)^3$ adds very few parameters to perform a multiplicative operation on the activations. Therefore, the loss landscape is not as smooth as for good zershot models, and due to this IA3 has to scale different activations in a very different manner to learn the task. Hence, compressing $(IA)^3$ to sparse sign-vector and a constant is not feasible. Whereas, In the case of Lora the updates are added and hence their impact on the final value of the parameter is not huge as the maximum of the LoRA parameter is still very small compared to the base model's parameter value.

Table 11: We present the performance$_{\text{(Storage Size in GB)}}$ on MMLU Validation for the compressed QLoRA models.

| Dataset (↓) | ComPEFT | | | |
|---|---|---|---|---|
| | 7B | 13B | 33B | 65B |
| **Self-Instruct** | 35.62 | 47.52 | 55.11 | 62.13 |
| **Longform** | 31.89 | 47.80 | 55.31 | 62.27 |
| **Chip2** | 33.49 | 47.15 | 55.02 | 62.21 |
| **HH-RLHF** | 32.37 | 47.19 | 54.78 | 62.06 |
| **Unnatural Instruct** | 42.41 | 49.62 | 56.28 | 62.15 |
| **Guanaco** | 33.92 | 49.52 | 55.35 | 62.00 |
| **Alpaca** | 39.82 | 49.00 | 55.91 | 62.37 |
| **FLAN v2** | 43.93 | 50.86 | 56.97 | 63.77 |
| **Average** | 37.88 | 48.58 | 55.59 | 62.37 |

Table 12: Validation set performance$_{\text{(Storage Size in MB)}}$ averaged over seven GLUE tasks when compressing $(\text{IA})^3$ and LoRA modules on different base models.

| Method (↓) | T5 − Base | | T5 − Large | | T0 − 3B | |
|---|---|---|---|---|---|---|
| | $(\text{IA})^3$ | LoRA | $(\text{IA})^3$ | LoRA | $(\text{IA})^3$ | LoRA |
| Original | 81.25 | 81.94 | 85.08 | 86.21 | 87.71 | 89.94 |
| ComPEFT | 81.04 | 80.96 | 85.28 | 86.54 | 89.14 | 89.95 |
| Improvement | -0.21 | -0.98 | 0.2 | 0.33 | 1.43 | 0.01 |

Table 13: **Task level results:** Average performance over 5 seed for LoraHub and ComPEFT for compositional generalization on Big-Bench-Hard.

| Task | Zeroshot | ICL | LoraHub (Avg) | ComPEFT (Avg) | LoraHub (Best) | ComPEFT (Best) |
|---|---|---|---|---|---|---|
| **Logical Deduction Three Objects** | 0.0 | 51.3 | 41.9 | 28.4 | 51.3 | 48.0 |
| **Tracking Shuffled Objects Five Objects** | 12.0 | 12.0 | 9.6 | 11.3 | 12.0 | 12.0 |
| **Web Of Lies** | 54.0 | 54.0 | 28.1 | 41.7 | 49.3 | 56.0 |
| **Tracking Shuffled Objects Seven Objects** | 6.7 | 6.7 | 5.3 | 6.7 | 6.7 | 6.7 |
| **Date Understanding** | 15.3 | 22.7 | 39.5 | 29.1 | 42.0 | 38.7 |
| **Navigate** | 47.3 | 44.0 | 48.4 | 38.5 | 50.7 | 50.0 |
| **Multistep Arithmetic Two** | 0.7 | 0.7 | 0.7 | 0.5 | 1.3 | 0.7 |
| **Boolean Expressions** | 54.0 | 58.7 | 55.9 | 55.7 | 57.3 | 61.3 |
| **Hyperbaton** | 6.7 | 74.0 | 55.2 | 49.9 | 65.3 | 67.3 |
| **Tracking Shuffled Objects Three Objects** | 24.7 | 30.7 | 26.7 | 21.6 | 29.3 | 24.7 |
| **Sports Understanding** | 56.0 | 56.0 | 46.4 | 53.1 | 54.7 | 58.0 |
| **Logical Deduction Seven Objects** | 12.7 | 42.0 | 35.5 | 37.6 | 40.0 | 40.0 |
| **Causal Judgement** | 57.5 | 56.3 | 40.7 | 49.2 | 58.6 | 57.5 |
| **Penguins In A Table** | 43.5 | 39.1 | 36.1 | 44.3 | 45.7 | 47.8 |
| **Geometric Shapes** | 6.7 | 18.7 | 9.6 | 7.3 | 19.3 | 9.3 |
| **Reasoning About Colored Objects** | 32.0 | 38.7 | 38.0 | 40.8 | 39.3 | 44.0 |
| **Dyck Languages** | 1.3 | 2.7 | 1.1 | 0.7 | 1.3 | 1.3 |
| **Disambiguation Qa** | 0.0 | 69.3 | 14.3 | 6.5 | 51.3 | 29.3 |
| **Salient Translation Error Detection** | 37.3 | 46.0 | 31.3 | 38.5 | 44.7 | 43.3 |
| **Movie Recommendation** | 62.7 | 52.7 | 61.1 | 58.0 | 67.3 | 62.0 |
| **Snarks** | 50.0 | 55.1 | 49.2 | 50.0 | 50.0 | 50.0 |
| **Formal Fallacies** | 51.3 | 58.0 | 41.3 | 41.1 | 52.7 | 51.3 |
| **Logical Deduction Five Objects** | 21.3 | 40.0 | 33.6 | 36.3 | 36.7 | 42.0 |
| **Temporal Sequences** | 16.7 | 26.7 | 18.7 | 19.5 | 20.0 | 21.3 |
| **Word Sorting** | 1.3 | 0.7 | 1.2 | 1.3 | 1.3 | 1.3 |
| **Ruin Names** | 23.3 | 18.7 | 18.0 | 22.4 | 23.3 | 23.3 |
| **Object Counting** | 34.7 | 32.0 | 35.5 | 35.3 | 36.7 | 36.0 |
| **Average** | 27.0 | 37.3 | 30.5 | 30.6 | 37.3 | 36.4 |

Table 14: Validation and Test set performance along with storage size in MB for bert-base-uncased Model, for $(IA)^3$, LoRA and Full model finetuning.

| PEFT | Task | Original (Val) | Original (Test) | ComPEFT (Val) | ComPEFT (Test) |
|------|------|----------------|-----------------|---------------|----------------|
| **full** | **mnli** | 84.7 | $83.1_{(208.8)}$ | 84.3 | $82.6_{(12.0)}$ |
| | **mrpc** | 86.8 | $97.5_{(208.8)}$ | 86.8 | $97.0_{(19.6)}$ |
| | **qnli** | 91.9 | $91.0_{(208.8)}$ | 91.9 | $90.9_{(12.0)}$ |
| | **qqp** | 90.1 | $90.3_{(208.8)}$ | 90.0 | $90.0_{(15.4)}$ |
| | **rte** | 66.4 | $94.5_{(208.8)}$ | 69.0 | $94.0_{(7.4)}$ |
| | **sst2** | 91.1 | $97.0_{(208.8)}$ | 92.2 | $96.0_{(7.4)}$ |
| | **wnli** | 56.3 | $57.0_{(208.8)}$ | 56.3 | $57.0_{(4.4)}$ |
| **ia3** | **mnli** | 79.2 | $78.9_{(0.1)}$ | 57.6 | $56.2_{(0.0)}$ |
| | **mrpc** | 84.6 | $94.5_{(0.1)}$ | 31.6 | $34.5_{(0.0)}$ |
| | **qnli** | 87.9 | $87.5_{(0.1)}$ | 49.3 | $49.6_{(0.0)}$ |
| | **qqp** | 84.6 | $84.4_{(0.1)}$ | 63.2 | $63.2_{(0.0)}$ |
| | **rte** | 59.2 | $73.5_{(0.1)}$ | 52.7 | $55.5_{(0.0)}$ |
| | **sst2** | 91.5 | $91.0_{(0.1)}$ | 49.1 | $45.5_{(0.0)}$ |
| | **wnli** | 54.9 | $57.0_{(0.1)}$ | 56.3 | $57.0_{(0.0)}$ |
| **lora** | **mnli** | 82.5 | $81.4_{(2.6)}$ | 76.6 | $76.9_{(0.2)}$ |
| | **mrpc** | 86.3 | $97.5_{(2.6)}$ | 82.8 | $94.0_{(0.2)}$ |
| | **qnli** | 91.7 | $91.0_{(2.6)}$ | 90.8 | $90.1_{(0.2)}$ |
| | **qqp** | 89.4 | $89.5_{(2.6)}$ | 87.4 | $87.5_{(0.2)}$ |
| | **rte** | 61.7 | $68.5_{(2.6)}$ | 58.8 | $66.0_{(0.2)}$ |
| | **sst2** | 92.4 | $92.5_{(2.6)}$ | 91.6 | $91.5_{(0.2)}$ |
| | **wnli** | 56.3 | $57.0_{(2.6)}$ | 59.2 | $56.0_{(0.1)}$ |

Table 15: Validation and Test set performance along with storage size in MB for bert-large-uncased Model, for $(IA)^3$, LoRA and Full model finetuning.

| PEFT | Task | Original (Val) | Original (Test) | ComPEFT (Val) | ComPEFT (Test) |
|------|------|----------------|-----------------|---------------|----------------|
| full | mnli | 85.4 | $84.0_{(639.2)}$ | 85.5 | $83.7_{(59.9)}$ |
|      | mrpc | 88.2 | $97.5_{(639.2)}$ | 88.5 | $97.5_{(47.2)}$ |
|      | qnli | 91.0 | $89.4_{(639.2)}$ | 91.2 | $89.6_{(36.8)}$ |
|      | qqp  | 88.6 | $88.5_{(639.2)}$ | 88.4 | $88.6_{(36.8)}$ |
|      | rte  | 71.5 | $94.0_{(639.2)}$ | 70.4 | $92.5_{(22.7)}$ |
|      | sst2 | 92.9 | $93.5_{(639.2)}$ | 93.5 | $92.5_{(36.8)}$ |
|      | wnli | 56.3 | $57.0_{(639.2)}$ | 57.8 | $58.0_{(13.4)}$ |
| ia3  | mnli | 82.4 | $81.9_{(0.3)}$ | 59.5 | $59.6_{(0.0)}$ |
|      | mrpc | 84.6 | $96.0_{(0.3)}$ | 31.6 | $34.5_{(0.0)}$ |
|      | qnli | 88.6 | $87.6_{(0.3)}$ | 59.6 | $59.9_{(0.0)}$ |
|      | qqp  | 87.7 | $87.4_{(0.3)}$ | 73.2 | $73.8_{(0.0)}$ |
|      | rte  | 58.8 | $72.5_{(0.3)}$ | 52.7 | $55.5_{(0.0)}$ |
|      | sst2 | 92.3 | $88.0_{(0.3)}$ | 49.1 | $45.5_{(0.0)}$ |
|      | wnli | 60.6 | $55.0_{(0.3)}$ | 56.3 | $57.0_{(0.0)}$ |
| lora | mnli | 83.6 | $82.9_{(6.8)}$ | 76.4 | $74.8_{(0.6)}$ |
|      | mrpc | 88.7 | $94.0_{(6.8)}$ | 87.0 | $92.0_{(0.5)}$ |
|      | qnli | 88.6 | $86.9_{(6.8)}$ | 82.6 | $81.1_{(0.6)}$ |
|      | qqp  | 87.0 | $87.1_{(6.8)}$ | 76.3 | $76.9_{(0.6)}$ |
|      | rte  | 59.9 | $77.0_{(6.8)}$ | 59.6 | $71.0_{(0.6)}$ |
|      | sst2 | 93.7 | $94.0_{(6.8)}$ | 93.6 | $93.0_{(0.4)}$ |
|      | wnli | 56.3 | $57.0_{(6.8)}$ | 57.8 | $56.0_{(0.2)}$ |

Table 16: Validation and Test set performance along with storage size in MB for roberta-base Model, for $(IA)^3$, LoRA and Full model finetuning.

| PEFT | Task | Original (Val) | Original (Test) | ComPEFT (Val) | ComPEFT (Test) |
|------|------|----------------|-----------------|---------------|----------------|
| full | mnli | 86.4 | $86.4_{(237.8)}$ | 86.6 | $86.2_{(13.7)}$ |
|      | mrpc | 87.0 | $89.0_{(237.8)}$ | 86.0 | $84.5_{(8.4)}$ |
|      | qnli | 91.8 | $91.2_{(237.8)}$ | 91.7 | $91.0_{(17.6)}$ |
|      | qqp  | 89.1 | $89.4_{(237.8)}$ | 89.1 | $89.2_{(17.6)}$ |
|      | rte  | 75.4 | $91.5_{(237.8)}$ | 78.0 | $93.0_{(22.3)}$ |
|      | sst2 | 95.2 | $95.2_{(237.8)}$ | 94.2 | $94.0_{(8.4)}$ |
|      | wnli | 56.3 | $56.0_{(237.8)}$ | 56.3 | $45.0_{(5.0)}$ |
| ia3  | mnli | 84.1 | $83.4_{(1.2)}$ | 43.0 | $44.7_{(0.1)}$ |
|      | mrpc | 88.7 | $98.0_{(1.2)}$ | 71.6 | $70.0_{(0.1)}$ |
|      | qnli | 89.7 | $88.9_{(1.2)}$ | 50.7 | $50.4_{(0.1)}$ |
|      | qqp  | 87.0 | $87.1_{(1.2)}$ | 80.9 | $80.8_{(0.1)}$ |
|      | rte  | 73.3 | $93.0_{(1.2)}$ | 54.9 | $54.5_{(0.1)}$ |
|      | sst2 | 93.5 | $92.0_{(1.2)}$ | 76.3 | $72.5_{(0.1)}$ |
|      | wnli | 56.3 | $57.0_{(1.2)}$ | 56.3 | $57.0_{(0.0)}$ |
| lora | mnli | 87.0 | $86.1_{(3.7)}$ | 86.2 | $85.6_{(0.3)}$ |
|      | mrpc | 89.5 | $98.5_{(3.7)}$ | 88.5 | $97.0_{(0.3)}$ |
|      | qnli | 91.1 | $92.3_{(3.7)}$ | 90.0 | $89.9_{(0.3)}$ |
|      | qqp  | 88.8 | $88.8_{(3.7)}$ | 88.2 | $88.4_{(0.3)}$ |
|      | rte  | 79.4 | $97.0_{(3.7)}$ | 79.4 | $96.0_{(0.3)}$ |
|      | sst2 | 94.2 | $95.0_{(3.7)}$ | 93.1 | $94.0_{(0.3)}$ |
|      | wnli | 56.3 | $57.0_{(3.7)}$ | 56.3 | $57.0_{(0.1)}$ |

Table 17: Validation and Test set performance along with storage size in MB for roberta-large Model, for $(IA)^3$, LoRA and Full model finetuning.

| PEFT | Task | Original (Val) | Original (Test) | ComPEFT (Val) | ComPEFT (Test) |
|------|------|----------------|-----------------|---------------|----------------|
| full | mnli | 90.6 | $89.3_{(677.8)}$ | 90.4 | $89.4_{(50.0)}$ |
|      | mrpc | 89.2 | $97.5_{(677.8)}$ | 89.2 | $97.5_{(24.1)}$ |
|      | qnli | 93.6 | $92.9_{(677.8)}$ | 93.5 | $93.3_{(63.5)}$ |
|      | qqp  | 90.1 | $89.9_{(677.8)}$ | 90.1 | $89.9_{(50.0)}$ |
|      | rte  | 85.6 | $98.5_{(677.8)}$ | 85.2 | $98.0_{(50.0)}$ |
|      | sst2 | 95.4 | $95.2_{(677.8)}$ | 96.3 | $95.0_{(50.0)}$ |
|      | wnli | 56.3 | $57.0_{(677.8)}$ | 57.8 | $61.0_{(63.5)}$ |
| ia3  | mnli | 89.5 | $88.5_{(2.3)}$ | 36.5 | $35.5_{(0.2)}$ |
|      | mrpc | 86.8 | $86.5_{(2.3)}$ | 68.4 | $65.5_{(0.0)}$ |
|      | qnli | 92.3 | $92.1_{(2.3)}$ | 51.6 | $50.4_{(0.1)}$ |
|      | qqp  | 88.5 | $87.9_{(2.3)}$ | 63.2 | $63.2_{(0.0)}$ |
|      | rte  | 80.1 | $94.0_{(2.3)}$ | 54.2 | $57.0_{(0.0)}$ |
|      | sst2 | 94.3 | $93.0_{(2.3)}$ | 55.2 | $59.5_{(0.2)}$ |
|      | wnli | 56.3 | $57.0_{(2.3)}$ | 60.6 | $51.0_{(0.1)}$ |
| lora | mnli | 89.9 | $89.3_{(8.8)}$ | 85.0 | $83.8_{(0.6)}$ |
|      | mrpc | 90.0 | $93.5_{(8.8)}$ | 90.4 | $90.0_{(0.8)}$ |
|      | qnli | 93.4 | $92.9_{(8.8)}$ | 91.0 | $90.3_{(0.8)}$ |
|      | qqp  | 89.1 | $88.9_{(8.8)}$ | 86.5 | $85.9_{(0.8)}$ |
|      | rte  | 80.5 | $93.5_{(8.8)}$ | 79.1 | $89.5_{(0.8)}$ |
|      | sst2 | 95.2 | $93.0_{(8.8)}$ | 94.8 | $90.5_{(0.8)}$ |
|      | wnli | 56.3 | $57.0_{(8.8)}$ | 56.3 | $57.0_{(0.2)}$ |

Table 18: Validation and Test set performance along with storage size in MB for t5-v1.1-base Model, for $(IA)^3$, LoRA and Full model finetuning.

| PEFT | Task | Original (Val) | Original (Test) | ComPEFT (Val) | ComPEFT (Test) |
|------|------|----------------|-----------------|---------------|----------------|
| full | mnli | 89.8 | $89.8_{(472.2)}$ | 88.8 | $89.0_{(34.9)}$ |
|      | mrpc | 80.9 | $74.5_{(472.2)}$ | 82.6 | $75.5_{(34.9)}$ |
|      | qnli | 88.0 | $88.6_{(472.2)}$ | 86.7 | $87.2_{(44.3)}$ |
|      | qqp  | 78.6 | $78.9_{(472.2)}$ | 77.0 | $77.6_{(27.2)}$ |
|      | rte  | 59.2 | $49.0_{(472.2)}$ | 59.2 | $61.0_{(9.9)}$ |
|      | sst2 | 93.4 | $91.0_{(472.2)}$ | 93.8 | $91.5_{(27.2)}$ |
|      | wnli | 56.3 | $47.0_{(472.2)}$ | 57.8 | $49.0_{(44.3)}$ |
| ia3  | mnli | 84.6 | $83.8_{(0.2)}$ | 54.3 | $54.4_{(0.0)}$ |
|      | mrpc | 82.8 | $81.5_{(0.2)}$ | 82.8 | $78.5_{(0.0)}$ |
|      | qnli | 85.2 | $86.3_{(0.2)}$ | 60.7 | $61.8_{(0.0)}$ |
|      | qqp  | 85.0 | $85.4_{(0.2)}$ | 78.6 | $78.8_{(0.0)}$ |
|      | rte  | 54.9 | $49.0_{(0.2)}$ | 63.2 | $62.5_{(0.0)}$ |
|      | sst2 | 92.3 | $91.0_{(0.2)}$ | 89.0 | $87.0_{(0.0)}$ |
|      | wnli | 52.1 | $57.0_{(0.2)}$ | 52.1 | $57.0_{(0.0)}$ |
| lora | mnli | 66.9 | $66.3_{(4.4)}$ | 56.6 | $57.1_{(0.2)}$ |
|      | mrpc | 72.1 | $67.0_{(4.4)}$ | 68.4 | $64.0_{(0.3)}$ |
|      | qnli | 87.1 | $88.8_{(4.4)}$ | 86.7 | $88.5_{(0.3)}$ |
|      | qqp  | 78.3 | $78.9_{(4.4)}$ | 72.0 | $72.2_{(0.2)}$ |
|      | rte  | 55.2 | $50.5_{(4.4)}$ | 53.1 | $49.5_{(0.2)}$ |
|      | sst2 | 93.0 | $91.5_{(4.4)}$ | 92.9 | $90.5_{(0.3)}$ |
|      | wnli | 56.3 | $47.0_{(4.4)}$ | 78.9 | $76.0_{(0.1)}$ |

Table 19: Validation and Test set performance along with storage size in MB for t5-v1.1-large Model, for $(IA)^3$, LoRA and Full model finetuning.

| PEFT | Task | Original (Val) | Original (Test) | ComPEFT (Val) | ComPEFT (Test) |
|------|------|---------------|-----------------|---------------|----------------|
| full | mrpc | 86.3 | $84.0_{(1493.7)}$ | 86.3 | $85.0_{(110.3)}$ |
|      | qnli | 94.0 | $94.0_{(1493.7)}$ | 94.4 | $94.9_{(140.0)}$ |
|      | qqp  | 90.2 | $90.5_{(1493.7)}$ | 89.4 | $89.5_{(140.0)}$ |
|      | rte  | 74.0 | $76.0_{(1493.7)}$ | 75.4 | $74.5_{(140.0)}$ |
|      | sst2 | 95.6 | $93.0_{(1493.7)}$ | 95.4 | $92.5_{(86.1)}$ |
|      | wnli | 52.1 | $57.0_{(1493.7)}$ | 52.1 | $57.0_{(31.4)}$ |
| ia3  | mnli | 92.0 | $92.4_{(0.5)}$ | 54.3 | $54.4_{(0.0)}$ |
|      | mrpc | 90.9 | $86.0_{(0.5)}$ | 77.9 | $77.5_{(0.0)}$ |
|      | qnli | 92.0 | $92.3_{(0.5)}$ | 79.4 | $78.1_{(0.0)}$ |
|      | qqp  | 87.2 | $87.5_{(0.5)}$ | 78.7 | $78.8_{(0.0)}$ |
|      | rte  | 69.7 | $67.0_{(0.5)}$ | 69.0 | $73.5_{(0.0)}$ |
|      | sst2 | 95.2 | $93.0_{(0.5)}$ | 79.5 | $81.0_{(0.0)}$ |
|      | wnli | 52.1 | $57.0_{(0.5)}$ | 52.1 | $57.0_{(0.0)}$ |
| lora | mnli | 92.3 | $93.2_{(11.8)}$ | 91.8 | $92.4_{(0.9)}$ |
|      | mrpc | 78.2 | $74.0_{(11.8)}$ | 77.7 | $76.0_{(0.9)}$ |
|      | qnli | 90.4 | $91.9_{(11.8)}$ | 87.0 | $87.4_{(0.9)}$ |
|      | qqp  | 87.1 | $87.6_{(11.8)}$ | 86.0 | $86.8_{(0.9)}$ |
|      | rte  | 52.7 | $56.0_{(11.8)}$ | 53.8 | $45.5_{(0.4)}$ |
|      | sst2 | 93.9 | $89.0_{(11.8)}$ | 61.4 | $56.0_{(0.4)}$ |
|      | wnli | 56.3 | $47.0_{(11.8)}$ | 56.3 | $47.0_{(0.4)}$ |

Table 20: Validation and Test set performance along with storage size in MB for t5-base Model, for $(IA)^3$, LoRA and Full model finetuning.

| PEFT | Task | Original (Val) | Original (Test) | ComPEFT (Val) | ComPEFT (Test) |
|---|---|---|---|---|---|
| **full** | **mnli** | 91.2 | $91.2_{(425.2)}$ | 89.9 | $90.4_{(31.4)}$ |
| | **mrpc** | 89.7 | $86.0_{(425.2)}$ | 87.0 | $75.5_{(39.9)}$ |
| | **qnli** | 93.3 | $93.3_{(425.2)}$ | 91.3 | $91.3_{(39.9)}$ |
| | **qqp** | 91.3 | $91.4_{(425.2)}$ | 70.4 | $70.6_{(15.1)}$ |
| | **rte** | 76.2 | $77.0_{(425.2)}$ | 74.7 | $77.0_{(39.9)}$ |
| | **sst2** | 95.6 | $93.5_{(425.2)}$ | 95.5 | $93.5_{(31.4)}$ |
| | **wnli** | 56.3 | $47.0_{(425.2)}$ | 56.3 | $48.0_{(24.5)}$ |
| **ia3** | **mnli** | 91.0 | $90.4_{(0.2)}$ | 90.4 | $90.3_{(0.0)}$ |
| | **mrpc** | 85.5 | $84.0_{(0.2)}$ | 85.8 | $81.5_{(0.0)}$ |
| | **qnli** | 92.6 | $92.9_{(0.2)}$ | 92.5 | $92.4_{(0.0)}$ |
| | **qqp** | 89.5 | $89.8_{(0.2)}$ | 87.1 | $87.0_{(0.0)}$ |
| | **rte** | 63.9 | $62.0_{(0.2)}$ | 65.0 | $58.5_{(0.0)}$ |
| | **sst2** | 94.2 | $93.0_{(0.2)}$ | 94.4 | $93.0_{(0.0)}$ |
| | **wnli** | 52.1 | $57.0_{(0.2)}$ | 52.1 | $57.0_{(0.0)}$ |
| **lora** | **mnli** | 91.0 | $90.2_{(6.2)}$ | 91.3 | $90.5_{(0.5)}$ |
| | **mrpc** | 90.9 | $84.0_{(6.2)}$ | 84.1 | $77.5_{(0.6)}$ |
| | **qnli** | 93.4 | $93.5_{(6.2)}$ | 93.3 | $93.7_{(0.5)}$ |
| | **qqp** | 90.5 | $90.5_{(6.2)}$ | 90.3 | $90.6_{(0.4)}$ |
| | **rte** | 52.7 | $53.0_{(6.2)}$ | 57.0 | $53.5_{(0.1)}$ |
| | **sst2** | 94.5 | $94.0_{(6.2)}$ | 94.4 | $93.5_{(0.4)}$ |
| | **wnli** | 60.6 | $49.0_{(6.2)}$ | 56.3 | $47.0_{(0.1)}$ |

Table 21: Validation and Test set performance along with storage size in MB for t5-large Model, for $(IA)^3$, LoRA and Full model finetuning.

| PEFT | Task | Original (Val) | Original (Test) | ComPEFT (Val) | ComPEFT (Test) |
|---|---|---|---|---|---|
| **full** | **mnli** | 93.4 | $93.6_{(1407.0)}$ | 93.1 | $93.2_{(81.1)}$ |
| | **mrpc** | 91.4 | $88.5_{(1407.0)}$ | 90.0 | $84.5_{(131.9)}$ |
| | **qnli** | 94.4 | $94.4_{(1407.0)}$ | 94.5 | $94.7_{(131.9)}$ |
| | **qqp** | 91.8 | $91.9_{(1407.0)}$ | 68.8 | $69.6_{(131.9)}$ |
| | **rte** | 83.8 | $88.0_{(1407.0)}$ | 82.0 | $82.0_{(131.9)}$ |
| | **sst2** | 93.5 | $93.0_{(1407.0)}$ | 93.6 | $93.0_{(131.9)}$ |
| | **wnli** | 56.3 | $47.0_{(1407.0)}$ | 78.9 | $76.0_{(103.9)}$ |
| **ia3** | **mnli** | 93.0 | $92.5_{(0.7)}$ | 93.0 | $92.7_{(0.1)}$ |
| | **mrpc** | 90.2 | $88.5_{(0.7)}$ | 90.7 | $90.0_{(0.0)}$ |
| | **qnli** | 94.4 | $94.2_{(0.7)}$ | 94.3 | $94.1_{(0.0)}$ |
| | **qqp** | 90.6 | $91.1_{(0.7)}$ | 89.3 | $90.0_{(0.0)}$ |
| | **rte** | 79.8 | $85.0_{(0.7)}$ | 82.0 | $83.5_{(0.0)}$ |
| | **sst2** | 95.5 | $95.0_{(0.7)}$ | 95.6 | $94.0_{(0.0)}$ |
| | **wnli** | 52.1 | $57.0_{(0.7)}$ | 52.1 | $57.0_{(0.0)}$ |
| **lora** | **mnli** | 93.0 | $93.5_{(16.5)}$ | 93.0 | $93.5_{(1.2)}$ |
| | **mrpc** | 90.9 | $87.5_{(16.5)}$ | 85.8 | $85.0_{(1.6)}$ |
| | **qnli** | 94.5 | $94.5_{(16.5)}$ | 94.1 | $92.9_{(1.6)}$ |
| | **qqp** | 90.9 | $91.4_{(16.5)}$ | 90.2 | $90.9_{(1.6)}$ |
| | **rte** | 82.0 | $82.0_{(16.5)}$ | 78.0 | $79.0_{(1.6)}$ |
| | **sst2** | 95.9 | $95.5_{(16.5)}$ | 95.8 | $94.0_{(1.6)}$ |
| | **wnli** | 56.3 | $47.0_{(16.5)}$ | 69.0 | $57.0_{(0.6)}$ |

Table 22: Validation and Test set performance along with storage size in MB for T0-3B Model, for $(IA)^3$, and LoRA.

| PEFT | Task | Original (Val) | Original (Test) | ComPEFT (Val) | ComPEFT (Test) |
|---|---|---|---|---|---|
| **ia3** | **mnli** | 94.1 | $94.4_{(1.0)}$ | 93.4 | $93.8_{(0.1)}$ |
| | **mrpc** | 89.7 | $89.5_{(1.0)}$ | 90.4 | $89.0_{(0.1)}$ |
| | **qnli** | 94.9 | $95.3_{(1.0)}$ | 95.8 | $95.5_{(0.0)}$ |
| | **qqp** | 89.8 | $90.2_{(1.0)}$ | 89.6 | $90.0_{(0.1)}$ |
| | **rte** | 86.6 | $89.0_{(1.0)}$ | 87.4 | $88.0_{(0.0)}$ |
| | **sst2** | 96.8 | $93.0_{(1.0)}$ | 96.9 | $93.0_{(0.0)}$ |
| | **wnli** | 62.0 | $74.0_{(1.0)}$ | 70.4 | $69.0_{(0.0)}$ |
| **lora** | **mnli** | 93.8 | $93.6_{(33.8)}$ | 93.5 | $94.2_{(2.5)}$ |
| | **mrpc** | 90.4 | $90.5_{(33.8)}$ | 90.0 | $88.5_{(1.9)}$ |
| | **qnli** | 95.8 | $94.7_{(33.8)}$ | 95.8 | $96.0_{(2.5)}$ |
| | **qqp** | 90.3 | $90.7_{(33.8)}$ | 90.4 | $90.8_{(3.2)}$ |
| | **rte** | 89.2 | $89.1_{(33.8)}$ | 88.4 | $90.0_{(2.5)}$ |
| | **sst2** | 96.8 | $95.0_{(33.8)}$ | 96.9 | $93.0_{(2.5)}$ |
| | **wnli** | 73.2 | $73.0_{(33.8)}$ | 74.6 | $74.0_{(3.2)}$ |

