# OpenReview forum: "ComPEFT: Compression for Communicating Parameter Efficient Updates via Sparsification and Quantization"
_TMLR — Accepted by TMLR_

### Review · Reviewer_XZfS · 2025-02-28

**Summary Of Contributions:**

This paper proposes a general compression framework for PEFT adapters (or model updates) based on sparsification followed by magnitude quantization. The method compresses model updates into a single scalar magnitude, with each parameter retaining only its sign (-1, 0, or +1), resulting in a sparse ternary representation. While the core technique is not novel, the paper introduces a variant that incorporates a scaling factor as a hyperparameter and conducts an extensive range of experiments to evaluate its effectiveness.

**Audience:**

Yes

**Broader Impact Concerns:**

No.

**Claims And Evidence:**

Yes

**Requested Changes:**

- Include experiments comparing the method to basic baselines, such as rank reduction.
- Discuss [1] in greater detail and add comparative experiments if necessary.

[1] Brüel-Gabrielsson, Rickard, et al. "Compress then serve: Serving thousands of lora adapters with little overhead." *arXiv preprint arXiv:2407.00066* (2024).

**Strengths And Weaknesses:**

- Strengths
    - **[Major]** A simple yet effective method, supported by extensive experimental evidence.
    - **[Major]** The paper is well-written and clearly presented.
    - **[Major]** Comprehensive experiments demonstrate the method’s applicability across various tasks, along with ablation studies that provide insights into the effects of hyperparameters.
- Weaknesses
    - I am uncertain whether a 3.2GB QLoRA adapter for LLaMA-65B is a good example. To achieve this size, the LoRA adapter would require approximately rank ~128 (please correct me if I am mistaken), resulting in around 1.6B parameters. This is relatively rare, and in such cases, I wonder how much the rank can be reduced while maintaining good performance. Can it also achieve a 26× storage reduction? My conjecture is that it likely cannot, but 5× is highly possible, and at least a discussion is needed. This also relates to the observation on page 3: *"a surprising finding that as the base model gets bigger, their task vectors become more compressible..."* In fact, Table 1 of [this paper](https://arxiv.org/pdf/2310.17513) already indicates that larger models require a lower LoRA rank to achieve the desired performance. Additionally, empirical evidence suggests that high-rank LoRA often underperforms low-rank LoRA in many scenarios, especially when the adapter is already overparameterized and the task is relatively simple. Therefore, it is unclear whether the observed benefits in compression and performance improvements stem from this specific method or just from the overparameterized LoRA adapter.
    - Some related works are not sufficiently discussed, particularly **"Compress then Serve: Serving Thousands of LoRA Adapters with Little Overhead."** Additionally, some fundamental baselines, such as rank reduction, are missing, as mentioned above.
    - **[Minor]** Typo in Algorithm 1: In the fourth line from the bottom, *tilde γ* should be used.
    - If I understand correctly, this method essentially performs sparse ternary compression (STE) but with an additional scaling factor, *α*. In Figure 5, STE appears significantly worse than the proposed method, but in Figure 6, the optimal *α* value is frequently 1. Could you clarify this?
    - In Appendix Table 10, the average performance of ComPEFT appears to be lower than ICL.

---

> ### Author Response · Authors · 2025-03-25
>
> We thank the reviewer for their time, effort, and for recognizing the strengths of our work and robust experiments. We also apologize for the delay in response due to some unforeseen circumstances.
>
> **Weakness-1 and Requested Change-1/2:** 3.2GB QLoRA adapter example. This is relatively rare, and in such cases, I wonder how much the rank can be reduced while maintaining good performance….
>
> **Response:** The example, 3.2GB lora is taken from the QLoRA checkpoints (for example: https://huggingface.co/timdettmers/guanaco-65b/tree/main), which are rank 64 LoRA adaptation on Llama-65B stored in fp32. The main point is to put in context that LoRA checkpoints for large models can be huge in themselves.
>
> We perform additional experiments to address the concern on rank-reduction baseline and the compression that can be achieved on smaller rank. We opt for the experimental setting from Table-2 where we work with the Llama-2 70B model. We perform experiments with rank 32 and 8 the results of which are attached below along with storage sizes.
>
> |Dataset|Lora64|ComLora64|Lora32|ComLora32|Lora8|ComLora8|
> |-|-|-|-|-|-|-|
> |alpaca-clean|67.13|67.56|66.98|67.24|64.82|65.27|
> |chip2|65.18|67|65.24|66.75|63.35|65.18|
> |longform|67.63|68.5|67.14|68.12|65.16|66.74|
> |oasst1|66.89|67.39|65.42|66.92|64.21|65.56|
> |self-instruct|62.36|67.18|62.68|67.48|61.32|65.81|
> |Average|65.84|67.53|65.49|67.30|63.77|65.71|
> |Size|1.58GB|56MB|790MB|28MB|197MB|7MB|
>
>
> From these results, we can see that; (1) at rank 32 there is a slight drop in performance compared to rank 64. We can compress the rank checkpoint as well by >25x (2) At rank 8, we see a significant drop in performance from 65.84 to 63.77. Moreover, ComPEFT can compress rank 8 lora as well by >25x. (3) for both rank 32 and 8, ComLoRA performs better than the original checkpoints. These results help us to conclude that the observed benefits in compression and performance improvements stem from ComPEFT as opposed to the overparameterized LoRA adapter.
>
> ---
>
> **Weakness-2:** Some related works are not sufficiently discussed ...
>
> **Response:** We will add a discussion in the related work section.
>
> ---
>
> **Weakness-3:** [Minor] Typo in Algorithm 1: In the fourth line from the bottom, tilde γ should be used.
>
> **Response:** The input to the reset_topk function should be the original task vector which is broken as $\gamma \odot \mu$, so the input should be $\gamma$. Please let me know if I am missing something here.
>
> ---
>
> **Weakness-4:** If I understand correctly, this method essentially performs sparse ternary compression (STE) but with an additional scaling factor, α. In Figure 5, STE appears significantly worse than the proposed method, but in Figure 6, the optimal α value is frequently 1. Could you clarify this?
>
> **Response:** As mentioned in Section 4.1 experimental setup, STC also uses an $\alpha$ equal to the mean of the magnitude across all the parameters. Hence, the setting of $\alpha=1$ is not similar to the STC method.
>
> ---
>
> **Weakness-5:** In Appendix Table 10, the average performance of ComPEFT appears to be lower than ICL.
>
> **Response:** Yes, Lorahub, a popular method for compositional generalization is not able to beat the ICL baseline. However, our main goal from the experiments is to show that the compressed checkpoint does not impact the performance during compositional generalisation and demonstrating that compeft checkpoints perform similar to the original ones is enough to make this claim.
>
> ---
>
> **Please let us know if you have any other questions and concerns. We will respond to them promptly. Moreover, we will incorporate all the feedback from our discussions along with the additional results at appropriate places in the paper in a more polished manner upon acceptance. Thanks!**

---

> ### Author Response · Authors · 2025-04-08
>
> Dear Reviewer XZfS,
>
> We greatly appreciate your thoughtful review and constructive suggestions for our manuscript. We wanted to kindly check if you require any additional information from us.
>
> Thank you so much for your time and consideration!
>
> Warmest regards,
> Authors

---

> ### Comment · Reviewer_XZfS · 2025-04-09
>
> Thanks so much. The newly added experiment addressed the most important concern I had.

---

### Review · Reviewer_GSQx · 2025-03-10

**Summary Of Contributions:**

The paper introduces ComPEFT, a compression technique for Parameter-Efficient Fine-Tuning (PEFT) that reduces communication and memory bottlenecks in serving expert modules for large language models. ComPEFT reduces PEFT module sizes by 8x - 50x without additional training while preserving or even improving performance.

**Audience:**

Yes

**Broader Impact Concerns:**

No concerns.

**Claims And Evidence:**

Yes

**Requested Changes:**

- **Expand Figure 3 to Include Comparisons**: Add HiRA, DoRA, VeRA, or other recent PEFT methods to Figure 3, along with their corresponding ComPEFT-compressed versions, to better demonstrate the method’s applicability and performance across different PEFT strategies.

- The paper does not seem to specify the rank settings used for LoRA and other PEFT methods in experiments. Please provide the details of these settings to ensure clarity and reproducibility.

**Strengths And Weaknesses:**

Summary of strengths

- ComPEFT leverages **sparsification and ternary quantization** to significantly reduce PEFT module size **without requiring additional training** while preserving or even enhancing model performance.
- ComPEFT **generalizes well across multiple model architectures** and demonstrates improved efficiency as model size increases, making it particularly effective for larger models.
- ComPEFT also improves **model merging performance** with minimized storage and communication overhead, enabling more efficient expert model serving.
- The paper provides a detailed and comprehensive analysis with ablations, comparisons, and practical deployment scenarios.

Summary of weaknesses

- The approach is primarily based on **magnitude-based pruning**, a well-explored model compression technique. While effective, its novelty is somewhat limited. The authors are encouraged to elaborate on the key differences between ComPEFT and prior works in this area.

- Lack of Comparisons with Advanced PEFT Methods: The experiments mainly focus on QLoRA, IA3, and LoRA, but do not include comparisons with more advanced PEFT approaches such as HiRA, DoRA, and VeRA, which have demonstrated further parameter reductions in LoRA. A comparison with these state-of-the-art techniques would strengthen the paper’s contributions.

  [1] HiRA: Parameter-Efficient Hadamard High-Rank Adaptation for Large Language Models (ICLR 2025 Oral)

  [2] DoRA: Weight-Decomposed Low-Rank Adaptation (ICML 2024 Oral)

  [3] VeRA: Vector-based Random Matrix Adaptation (ICLR 2024)

---

> ### Author Response · Authors · 2025-03-25
>
> We thank the reviewer for their time, effort, and for recognizing the strengths of our work and robust experiments. We also apologize for the delay in response due to some unforeseen circumstances.
>
> **Weakness-1:** The approach is primarily based on magnitude-based pruning, a well-explored model compression technique...
>
> **Response:** There differences between the general pruning literature and ComPEFT are: (1) most of the pruning literature focuses on compressing the model weight. However, the focus of ComPEFT and other similar methods DAREx and BitDelta, is on compressing the parameter updates (also known as task vectors or delta vectors as mentioned in Introduction paragraph-3). These updates show very different compression behaviors compared to full model parameters. When compressing full model parameters, we start to see performance drop as the pruning ratio becomes high (pruning more than 50% params), in contrast, task vectors can be pruned by 90% without dropping the performance much. Additionally, when pruning full model weight, the performance drop is typically recovered by performing additional retraining, while for task vectors we can recover the mild drop by rescaling the pruned task vectors such that it ensures activation norms similar to the original model. Recently, task vector compression methods like DAREx and BitDelta focus on pruning and binary quantization respectively. In contrast, ComPEFT combines both of these ideas to remove noise and compresses the task vectors to extreme levels while maintaining performance.
>
> Hence, the novelty of our work, similar to DAREx (ICLR’25) and BitDelta (NeurIPS’24) lies in showing the applicability of these ideas for some practical use cases. ComPEFT has been built keeping in mind the emerging field of decentralized training, model merging, post-hoc mixture of expert routing moerging systems, and large scale efficient serving of PEFT modules where the parameter update needs to be swapped/communicated. Moreover, the novelty of ComPEFT also lies in the combination of using both Sparsity and Quantization in a simple and coherent manner that yields good performance while leading to very high compression. In Contrast, DAREx only uses pruning, while BitDelta relies only on quantization
>
> ---
>
> **Weakness-2 and Requested Change-1:** Lack of Comparisons with Advanced PEFT Methods: The experiments mainly focus on QLoRA, IA3, and LoRA, but do not include comparisons with more advanced PEFT approaches such as HiRA, DoRA, and VeRA, which have demonstrated further parameter reductions in LoRA...
>
> **Response:** ComPEFT has been built keeping in mind the emerging field of decentralized training, model merging, post-hoc mixture of expert routing moerging systems, and large scale efficient serving of thousands Lora modules where the parameter update needs to be swapped/communicated. Given this our primary focus is on post-hoc compression of popular PEFT methods like Lora and IA3. For example, Hugginface hub has 57,319 LoRa models, while only 887 DoRA models, 144 VeRA models, and almost no HiRA based models. Hence, the main focus is on the LoRA model while IA3 serves as a proof of concept that the ideas extend to different peft methods.
>
> However, to address your concerns we conducted some experiments. For the experimental setting in Table-2 with rank 64 LoRA on the Llama-2 70B model. We performed additional experiments with DoRA of rank 64 and then compressed them using ComPEFT and reported the results. We omitted VeRA methods as based on the DoRA paper VeRA typically performs worse than both LoRA and DoRA. Lastly, we omitted HiRA as the method as due to its recency its code is not available.
>
> |Dataset|LORA|ComLORA|DORA|ComDORA|
> |---|---|---|---|---|
> |alpaca-clean|67.13|67.56|68.42|69.78|
> |chip2|65.18|67|67.21|68.32|
> |longform|67.63|68.5|69.36|68.92|
> |oasst1|66.89|67.39|68.89|67.63|
> |self-instruct|62.36|67.18|65.26|66.31|
> |Average|65.84|67.53|67.83|68.19|
> |Size|1.58GB|56MB|1.59GB|57MB|
>
>
> Similar to our other finding, we observe that ComPEFT can also compress DoRA checkpoints to a great extent while preserving performance. Moreover, ComDoRA checkpoints slightly outperform ComLoRA’s performance.
>
> ---
>
> **Requested Changes-2:** The paper does not seem to specify the rank settings used for LoRA and other PEFT methods in experiments. Please provide the details of these settings to ensure clarity and reproducibility.
>
> **Response:** For Llama models, we follow the QLoRA paper and use a rank of 64, while for the other models we use rank=8. We will add this along with some other information to the main paper.
>
> ---
>
> **Please let us know if you have any other questions and concerns. We will respond to them promptly. Moreover, we will incorporate all the feedback from our discussions along with the additional results at appropriate places in the paper in a more polished manner upon acceptance. Thanks!**

---

> ### Author Response · Authors · 2025-04-08
>
> Dear Reviewer GSQx,
>
> We greatly appreciate your thoughtful review and constructive suggestions for our manuscript. We wanted to kindly check if you require any additional information from us.
>
> Thank you so much for your time and consideration!
>
> Warmest regards,
> Authors

---

> > ### Comment · Reviewer_GSQx · 2025-04-09
> >
> > I appreciate the author's responses, and most of my concerns have been addressed. I recommend accepting this paper.

---

### Review · Reviewer_Jdms · 2025-03-10

**Summary Of Contributions:**

This paper introduces ComPEFT, a compression method for delta parameters that significantly reduces their storage and communication overhead without requiring additional training. The core idea is to leverage sparsity and quantization to achieve high compression rates while maintaining or even improving model performance.

**Audience:**

Yes

**Broader Impact Concerns:**

No concern

**Claims And Evidence:**

Yes

**Requested Changes:**

1. Add discussion on delta compression methods [1,2]
2. Adding comparisons with   [1,2] would strengthen the paper. Even if the combination lacks novelty, demonstrating superior performance would still make it valuable.

[1] DARE the Extreme: Revisiting Delta-Parameter Pruning For Fine-Tuned Models, ICLR 2025.

[2] BitDelta: Your Fine-Tune May Only Be Worth One Bit, NeurIPS 2024.

**Strengths And Weaknesses:**

**Strengths:**
1. The paper addresses delta parameter compression, a crucial topic for improving the efficiency of large language models (LLMs).

2. It presents experiments across different model scales and tasks, demonstrating robustness.

3. The proposed ComPEFT method effectively reduces storage requirements, making PEFT more practical for deployment.

**Weaknesses:**

1. The paper lacks references to existing methods. For example, DAREx [1] efficiently prunes over 90% of delta parameters, and BitDelta [2] compresses delta parameters into 1-bit representations. A discussion of these approaches would provide better context.

2. Limited novelty: The method appears to be a combination of delta pruning and compression, both of which have been explored in prior work [1,2].

3. Limited benchmarking against delta compression methods: While the paper compares ComPEFT to STC and baseline PEFT techniques, it does not evaluate its performance against delta compression methods such as [1,2]. Adding these comparisons would strengthen the paper. Even if the combination lacks novelty, demonstrating superior performance would still make it valuable.

[1] DARE the Extreme: Revisiting Delta-Parameter Pruning For Fine-Tuned Models, ICLR 2025.

[2] BitDelta: Your Fine-Tune May Only Be Worth One Bit, NeurIPS 2024.

---

> ### Author Response · Authors · 2025-03-25
>
> We thank the reviewer for their time, effort, and for recognising the strengths of our work and robust experiments. We also apologize for the delay in response due to some unforeseen circumstances.
>
> **Weakness-1/3 and Requested Change-1/2:** Lacks references and comparison with DAREx and BitDelta.
>
> **Response:** First of all we would like to note that DAREx is a very recent work, however, to address your concerns we have included comparison with the suggested methods below. Based on our discussion, we promise to add these results in a more polished manner to the main paper upon acceptance.
>
> We performed experiments in a setting similar to Table-2, where we worked with the Llama-2 70B model and learned qLora modules of rank 64. We then compressed these parameter updates using ComPEFT, STC, BitDelta, and DAREx methods. Note that the BitDelta method has two variants. The first variant does not perform any additional training for the scale parameter (referred to as “No Training”). In the “BitDelta (No Training)” setting, the scale parameter ($\alpha$) is set to the mean value of all the parameters in the task vector/delta weights. The “BitDelta (Training)” variant learns the scale parameter ($\alpha$) via SGD and hence is not directly comparable with our ComPEFT which requires no additional training. For the DAREx method we use the DAREx-q (1/**q_v** ) variant, which uses labelled data to select the inverse scaling parameter (**q_v**) for each per-layer separately after pruning. We DAREx, we use sparsity levels of 95% and 99% as used in their paper. The results for the experiments are provided below along the average sizes of the compressed parameters across all the tasks.
>
>
> |Dataset|Original|ComPEFT|STC|BitDelta(NoTraining)|BitDelta(Training)|DAREx-q_v(p=0.95)|DAREx-q_v(p=0.99)|
> |-|-|-|-|-|-|-|-|
> |alpaca-clean|67.13|67.56|66.57|66.27|67.43|65.85|39.57|
> |chip2|65.18|67|64.54|64.31|67.31|63.94|50.18|
> |longform|67.63|68.5|67.02|66.15|68.61|66.14|44.32|
> |oasst1|66.89|67.39|66.15|65.38|67.11|65.48|45.82|
> |self-instruct|62.36|67.18|61.94|61.52|66.82|61.97|49.39|
> |Average|65.84|67.53|65.24|64.73|67.46|64.68|45.86|
> |Size|1.58GB|56MB|56MB|99MB|99MB|395MB|79MB|
>
>
> From these results, we can clearly see that: (1) ComPEFT performs better than these baseline. (2) DAREx (p=0.95) and BitDelta(No Training) show slight performance loss compared to the original checkpoint while DAREx (p=0.99) results in a huge drop. This is in line with the results presented in their papers. (3) BitDelta (Training) performs similar to ComPEFT, however, this method learns the scalar ($\alpha$) which requires both forward and backward passes and hence more GPU memory. (4) Note that BitDelta (No Training) sets the scalar ($\alpha$) as the mean of all the values in the task vector. It is very similar to STC which also uses the mean value as the scalar. However, they have a critical difference which is that STC also performs sparsification before performing quantization. Hence, in BitDelta the values are (+a, -a) while in STC the values are like (+b,0,-b). We note that STC performs slightly better than BitDelta (No Training), we believe that this is due to the sparsification step which removes redundant parameters which add noise. Similar phenomenon is also observed in [1, TIES-Merging]. Lastly, we also report the storage size for the compressed checkpoints where we use different methods to store them. We use golomb coding for ComPEFT/STC, bitmask for Bitdelta, and coo_sparse matrix for DAREx method. The results demonstrate that ComPEFT yields better performance/size trade-off compared to most of these other methods.
>
> ---
>
> **Weakness-2:** Limited novelty: The method appears to be a combination of delta pruning and compression, both of which have been explored in prior work [1,2].
>
> **Response:** The idea of magnitude/random pruning has existed throughout the pruning literature and similar things is true for quantization where people use binary and ternary quantization for efficiency and also for communicating gradients. Hence, the novelty of our work, similar to DAREx and BitDelta lies in showing the applicability of these ideas for some practical use cases. ComPEFT has been built keeping in mind the emerging field of decentralized training, model merging, post-hoc mixture of expert routing moerging systems, and large scale efficient serving of PEFT modules where the parameter update needs to be swapped/communicated. Moreover, the novelty of ComPEFT also lies in the combination of using both Sparsity and Quantization in a simple and coherent manner that yields good performance. In Contrast, DAREx only uses pruning, while BitDelta relies only on quantization.
>
> ---
>
> **Please let us know if you have any other questions and concerns. We will respond to them promptly. Moreover, we will incorporate all the feedback from our discussions along with the additional results at appropriate places in the paper in a more polished manner upon acceptance. Thanks!**

---

> ### Comment · Reviewer_Jdms · 2025-03-25
>
> I appreciate the author's additional experiments. However, I have some disagreements regarding the paper's novelty on that this paper is equivalent to BitDelta and DAREx.
>
> 1. BitDelta: This method was the **first** to demonstrate that quantization can be used for delta parameter compression and provided a high-level explanation of why this approach is effective. Additionally, it requires minimal GPU memory since it only stores gradients for the alpha parameter, making training lightweight. As stated in their paper, BitDelta requires **only 200 training steps**. Although performance appears similar to BitDelta, but it is okay as the memory size is reduced.
>
> 2. DAREx: This method focus on **random** dropping to enable **high flexibility** in pruning delta parameters and includes a comprehensive theoretical analysis explaining why random dropping is effective. As noted in their paper, it supports **structured pruning**, aligning well with hardware efficiency. Furthermore, this flexibility can be applied to better **mitigate interference** in model merging [1] while the importance-based pruning in this paper may face challenges when multiple tasks share crucial neurons.
>
> 3. It is nice to show that DAREx can perform well on 95\% without quantization. As show in your paper, your $k$ for importance pruning is maximum $50\%$.  Can you try to use the DAREx to replace the magnitude pruning in your methods to see how the performance being? This can further benefit your combination of the quantization and pruning as DAREx can do structured pruning and combined with quantization, we only need to calculate a small structured low bit matrix on hardware for efficiency.
>
> Overall I like the paper's idea on using both Sparsity and Quantization in a simple and coherent manner that yields good performance, but I think the author should provide a more clear discussion on BitDelta and DAREx to give audience a better understanding on how to choose the methods under their requirements.
>
> I will increase my score if the author resolving the additional issue I have.
>
> [1] TIES-MERGING: Resolving Interference When Merging Models

---

> > ### Author Response · Authors · 2025-03-26
> >
> > Thank for indulging in the discussion, we appreciate your time and efforts!
> >
> > **Response to Point-1:** Our results show that BitDelta (no training) variant what needs similar resources as ComPEFT is on average 2.8% worse than ComPEFT. Whereas, the BitDelta (training) variant has comparable performance but need to perform additional retraining to recover the performance from their quantization. Training to recover performance is prevalent throughout the compression literature focusing on full parameter compression. However, one of our main insight is that for parameter updates behave differently than full parameters and for compressing them we do not need to perform retraining which is resource intensive. We can get all the benefit of BitDelta while compressing more and needing no additional training. Hence, the benefits include no additional GPU memory for storing activations etc during training and reduced model size.
> >
> > ---
> >
> > **Response to Point-2 (Structured pruning for efficiency):** Based on our past experiments, changing the pruning method results in small changes in performance. However, due to the efficiency argument for structued pruning, we tried using structured DAREx pruning as opposed to importance based pruning in ComPEFT in a similar experimental setting to our previous results. We provide the results below. We observe that ComPEFT(DAREx_qv) is 1.39% worse on average but the performance is still higher than the original checkpoints. As we are not sure what the optimal way to store the structured pruned and quantized checkpoint, hence we omitted the sizes. We hope this addresses your concerns.
> >
> > |Dataset|Original|ComPEFT|DAREx-q_v(p=0.95)|ComPEFT(DAREx_qv,p=0.05)|
> > |-|-|-|-|-|
> > |alpaca-clean|67.13|67.56|65.85|66.32|
> > |chip2|65.18|67|63.94|65.78|
> > |longform|67.63|68.5|66.14|66.83|
> > |oasst1|66.89|67.39|65.48|67.53|
> > |self-instruct|62.36|67.18|61.97|64.25|
> > |Average|65.84|67.53|64.68|66.14|
> >
> > ---
> >
> > **Response to Point-2 (Better Model Merging):** In our Table-6, we perform merging experiment with ComPEFT compressed checkpoints and find that in 9/12 experimental setting ComPEFT checkpoint improve model merging. Hence, the importance-based pruning in this paper **does not** face challenges when merging models. Moreover, the original TIES-Merging paper, also used importance based pruning to resolve interference when merging models.
> >
> > ---
> >
> > **Response to Point-3:** As mentioned "section 2.1, sparsify", K denotes the percent of topk paramters to keep. In the paper, we explore k in the range {5,10,20,30,50} (as mentioned in page-6 line 1). The value of K=5 means we keep the top-5% of the parameters by magnitude and reset the rest to 0. Hence, K=5 is similar to p=0.95 for DAREx. For most of the experiments in the paper, the optimal value of K is 5 or 10 meaning 90%-95% sparsity. So the sparsity level used in ComPEFT are already comparable to the values used in DAREx.
> >
> > For the requested results please refer to **Response to Point-2 (Structured pruning for efficiency)**
> >
> > ---
> >
> > **We hope our explanations have successfully addressed the points raised regarding the utility of our method in comparison to previous work. We promise to update the final paper to position our work properly with respect to the prior work and provide guideline as when which method might be useful. We value the reviewer's feedback and hope they now see the benefits and potential contributions of our approach.**

---

> ### Comment · Reviewer_Jdms · 2025-03-26
>
> For the importance based delta pruning, I'm sure that it will suffer from high pruning rate  for most models, as shown in [1,2], maybe in your finetuned model they are similar.  But thank you for your response, my concerns have been resolved, I have increased my score
>
> [1]DARE the Extreme: Revisiting Delta-Parameter Pruning For Fine-Tuned Models, ICLR 2025.
> [2] Language Models are Super Mario: Absorbing Abilities from Homologous Models as a Free Lunch

---

> > ### Author Response · Authors · 2025-03-27
> >
> > Thank you for the suggestion which improve the quality of our paper. Thanks you for you time and help!

---

### Decision · Action_Editor_gSNR · 2025-04-11

**Recommendation:** Accept as is

**Comment:**

The paper presents ComPEFT, a practical compression method significantly reducing the size of PEFT modules without additional training. Reviewer discussions mainly concerned novelty, baseline comparisons, and compatibility with recent PEFT methods. The authors resolved these by conducting comprehensive experiments comparing ComPEFT against relevant prior work, including structured pruning and LoRA variants. Reviewers ultimately recognized the practical impact, thorough experiments, and effective clarifications provided by the authors, recommending acceptance with minor revisions to clearly incorporate these additional results into the final manuscript.

Overall, the paper addresses an important practical challenge and proposes a simple yet robust approach that enables efficient deployment of LLMs. By effectively compressing PEFT modules, ComPEFT enhances flexibility and efficiency during inference, significantly benefiting real-world applications involving a large number of experts and/or frequent expert module swapping.

**Audience:**

The paper is highly relevant and would interest the TMLR audience, particularly those researching PEFT and LLMs. Given the increasing importance of efficient deployment methods/systems for LLMs, ComPEFT's practical contributions and comprehensive experimental evaluations offer clear value to practitioners and researchers alike.

**Claims And Evidence:**

Yes. Their proposed method, ComPEFT, combines sparsification and quantization to compress model parameters involved with parameter-efficient fine-tuning (PEFT), and it is thoroughly evaluated across diverse models and tasks. Authors address reviewer concerns by performing extensive additional experiments, clearly demonstrating superior or comparable performance relative to state-of-the-art methods and showing that it is compatible with newer PEFT methods (e.g., LoRA variants). They also clarified points on novelty and baseline comparisons. The presented evidence convincingly validates the efficacy of ComPEFT in practical scenarios.